# Progressive release of long-stored carbon from tropical peatland disturbances

Jun Koarashi [1] ✉, Masayuki Itoh [2], Mariko Atarashi-Andoh[1], Yoko Saito-Kokubu [3,4], Makoto Matsueda [5], Kitso Kusin [6,7], Adi Jaya[6,7], Salampak Dohong [7] & Takashi Hirano [8]

Tropical peatlands are globally important, millennia-old carbon sinks, yet unprecedented human-driven degradation is triggering alarming carbon emissions. Comprehensive quantification of carbon dynamics across the disturbance sequence—from peat swamp forests to drained and fire-impacted peatlands—remains a critical knowledge gap. Here we show that over 18 years (1996–2014), drainage and subsequent peat fires released approximately 30–41 kg C m$^{-2}$ from peatlands in Central Kalimantan, Indonesia, using radiocarbon dating of peat profiles and groundwater dissolved organic carbon. Drainage contributed 5–11 kg C m$^{-2}$, primarily from centuries- to millennium-old, previously waterlogged peat. Fires released 23–32 kg C m$^{-2}$ from peat accumulated over the past 3,000 years, initiating progressive oxidative decomposition of older peat. Extrapolation to Indonesia's disturbed peatlands suggests a release of 0.81–3.70 Gt C between 1996 and 2014, with ongoing decomposition releasing an additional 0.03–0.08 Gt C annually, accelerating the impact on the global carbon balance.

Although they cover only 0.3% of Earth's land[1], tropical peatlands hold an estimated 105 Pg of carbon (C)[2]—more than 6% of the world's soil C stock (1700 Gt C)[3]. For millennia, these waterlogged, anaerobic ecosystems have quietly built up this C reservoir[1,2,4–6]. This vital stock is now threatened by human-driven disturbances, such as deforestation, drainage, fire, and agricultural conversion, transforming long-term C sinks into significant sources of C emissions[7–14]. While tropical peatlands in Africa[4] and Central[15]/South America[16] remain relatively intact, Southeast Asia, which contains the largest area (25 Mha)[1] of these ecosystems, has experienced extensive and accelerating peatland degradation[17,18].

Since the 1970s, rapid population growth and escalating demand for agricultural land and timber[2] have driven devastating logging and land development in Southeast Asian peatlands[19]. These logging and

development have resulted in a drastic shift in landscape composition, with secondary forests, plantations, and fire-prone shrublands replacing native peat swamp forests. Over the past three decades, native forests have dramatically decreased in area, with less than 40% (4.6 Mha) of native forests remaining undisturbed in peninsular Malaysia, Borneo, and Sumatra, whereas agricultural and abandoned lands have expanded significantly[18].

This land conversion triggers a cascade of detrimental effects. Deforestation and drainage accelerate oxidative peat decomposition[20], and subsequent agricultural expansion and exploitation produce the conditions for catastrophic peat fires[12,21]. These processes release accumulated C, primarily as carbon dioxide ($CO_2$), directly into the atmosphere, effectively reversing the long-term C sequestration function of these ecosystems[9,10]. In western Indonesia,

[1]Nucelar Science and Engineering Center, Japan Atomic Energy Agency, Ibaraki, Japan. [2]Research Institute for Sustainable Humanosphere, Kyoto University, Uji, Japan. [3]Tono Geoscience Center, Japan Atomic Energy Agency, Toki, Japan. [4]Integrated Support Center for Nuclear Nonproliferation, Security and Human Resource Development, Japan Atomic Energy Agency, Ibaraki, Japan. [5]Collaborative Laboratories for Advanced Decommissioning Science, Japan Atomic Energy Agency, Fukushima, Japan. [6]Center for International Cooperation in Sustainable Management of Tropical Peatland, University of Palangka Raya, Palangka Raya, Indonesia. [7]Department of Agriculture, University of Palangka Raya, Palangka Raya, Indonesia. [8]Research Faculty of Agriculture, Hokkaido University, Sapporo, Japan. ✉e-mail: koarashi.jun@jaea.go.jp

annual C losses due to drainage and fires are estimated to be 28-fold greater than predisturbance rates of carbon uptake[22]. Crucially, the release is not limited to recent C; ancient C, which has been locked within peat for thousands of years, can be lost from deeper peat layers in drained and burnt areas, particularly via fluvial pathways[8,23]. While episodic fires generate immediate, visible C releases[12], continuous oxidative peat decomposition following these events potentially leads to an alarming loss of this older, previously preserved peat C pool[24].

Despite the increasing recognition of this crisis, significant uncertainties persist in our understanding of C dynamics in peatlands. Although there are studies examining drainage and wildfire impacts on tropical peatlands[25–27], comprehensive assessments of the magnitude and age of C loss throughout the entire sequence of disturbances— from undrained peatlands to drainage and subsequent fire impacts— are still critically lacking[2,9,28].

Here, we quantify the impact of sequential disturbances on C stock and release in a representative peat landscape of Central Kalimantan, Indonesia, utilizing radiocarbon ($^{14}C$) dating of peat profiles and dissolved organic C (DOC). We examine three peatland sites that represent a sequence of disturbance within an edaphically similar area: an undrained swamp forest (UF), a drained forest (DF), and a drained, repeatedly burnt ex-forest (DB) (Table 1), and characterize peat C as a cumulative stock function of $^{14}C$ age (Fig. 1 and Supplementary Table 1) rather than peat depth. These three sites are located within 15 km on flat terrain around the edge of peat domes, originally exhibiting similar vegetation, peat thickness, and depth profiles of dry bulk density, C concentration, and $^{14}C$ age of the peat (Table 1, Fig. 1, Supplementary Method 1 and Supplementary Tables 1 and 2). A comparison of these characteristics across sites offers an opportunity to assess peat C preservation and loss due to disturbance regimes.

## Results and discussion

The drainage of peat swamp forests lowers the groundwater level (GWL), increases exposure of peat to oxygen, and stimulates aerobic microbial activity, thereby accelerating the decomposition of the peat C reservoir[2,13,14]. Our analysis reveals that during 18 years, drainage released approximately 5–11 kg C m$^{-2}$, primarily from the upper 45 cm of the peat layer that had accumulated over the last millennium (Fig. 1, Supplementary Method 2 and Supplementary Fig. 1). This drainage-induced peat C loss represents 21–47% of the existing C stock in the surface 45 cm (approximately 24 kg C m$^{-2}$ at the UF site) and is attributable to preferential decomposition of recalcitrant organic compounds, including aromatic-rich components (lignin and humic substances)[29,30] (Fig. 2), underscoring the critical impact of drainage. Given the nearly constant dry bulk densities and C concentrations in the peat profile at UF (Supplementary Table 1), the subsidence caused by this peat loss at DF is estimated at approximately 9–21 cm, with an annual rate of 0.5–1.2 cm y$^{-1}$. This range aligns with recent estimates for Southeast Asian drained peatlands[31]. The annual average drainage-induced C loss (0.27–0.62 kg C m$^{-2}$ y$^{-1}$) in our study closely matches a recent estimate of the difference in net ecosystem exchange between the UF and DF sites (0.41 kg C m$^{-2}$ y$^{-1}$, averaged over 20 years from 1996 to 2016)[10], suggesting that a substantial proportion of the observed increase in ecosystem-scale $CO_2$ emissions is driven by accelerated peat decomposition following drainage[20,32–35].

Carbon dioxide emission from the peat surface (gaseous C release) and organic C export via rivers (fluvial C release) represent the primary pathways for peat-C loss from drained peatlands[2,8,36]. Using a closed chamber method with root-trenching, Itoh et al.[20] measured $CO_2$ emissions from the peat surface at the UF and DF sites during the 2014 El Niño drought year, estimating annual $CO_2$ emissions through oxidative peat decomposition to be 698 and 775 g C m$^{-2}$ y$^{-1}$ for the UF and DF sites, respectively (Table 1). Moore et al.[8] independently monitored DOC and particulate organic C (POC) concentrations, along with water discharge rates, from channels draining areas around the

**Table 1 | Ecosystem C balances, $CO_2$ emissions from peat decomposition, groundwater levels, and peat thickness at three peatland sites**

| Site[a] | Location | NEE[b] (g C m$^{-2}$ y$^{-1}$) | GPP[b] (g C m$^{-2}$ y$^{-1}$) | RE[b] (g C m$^{-2}$ y$^{-1}$) | $CO_2$ emissions from peat decomposition[c] (g C m$^{-2}$ y$^{-1}$) | Mean annual GWL[b] (m) | Peat thickness[d] (m) |
|---|---|---|---|---|---|---|---|
| UF | 2.32°S, 113.90°E | −41 ± 371 | 3789 ± 179 | 3748 ± 268 | 698 | −0.15 ± 0.13 | 3.0 |
| DF | 2.35°S, 114.04°E | 353 ± 324 | 3181 ± 260 | 3534 ± 307 | 775 | −0.46 ± 0.13 | 4.5 |
| DB | 2.34°S, 114.04°E | −182 ± 429 | 1904 ± 250 | 1722 ± 320 | 646 | −0.17 ± 0.19 | 4.5 |

[a]UF: Undrained swamp forest; DF: Drained forest; DB: Drained, repeatedly burnt ex-forest.
[b]Average and standard deviation of net ecosystem $CO_2$ exchange (NEE), gross primary production (GPP), ecosystem respiration (RE), and groundwater level (GWL) from 2005 to 2016 (ref. 10).
[c]Values measured in 2014 (ref. 20).
[d]Ref. 20.

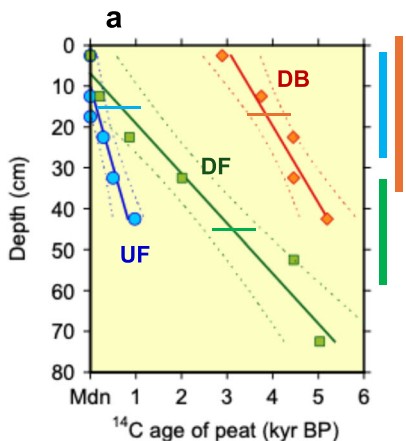

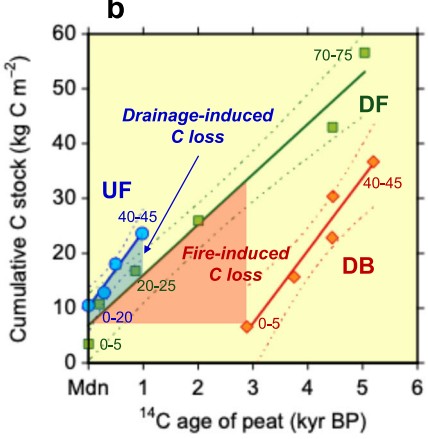

**Fig. 1 | Radiocarbon age and cumulative carbon stock of peat.** $^{14}$C age depth profiles of peat (**a**) and cumulative peat carbon stocks plotted against $^{14}$C age (**b**) for three peatland sites in Palangka Raya, Indonesia. UF: undrained swamp forest, DF: drained forest, and DB: drained, burnt ex-forest. The 95% confidence intervals (upper and lower bounds) are indicated by the dashed lines. Mdn on the *X*-axis denotes the modern $^{14}$C age (set to 0 kyr BP for calculations). Linear relationships are observed in both panels: (**a**) UF ($r = 0.93$, $p < 0.01$), DF ($r = 0.98$, $p < 0.001$), and DB

($r = 0.97$, $p < 0.01$); (**b**) UF ($r = 0.99$, $p < 0.05$), DF ($r = 0.99$, $p < 0.0005$), and DB ($r = 0.97$, $p < 0.01$). In (**a**), the mean annual groundwater level and its range at each site[10] are indicated by a horizontal bar within the figure and a vertical bar to the right of the figure, respectively. In (**b**), the peat depth (cm) is shown numerically, and the blue/red shading represents estimated C losses due to drainage/fires, respectively.

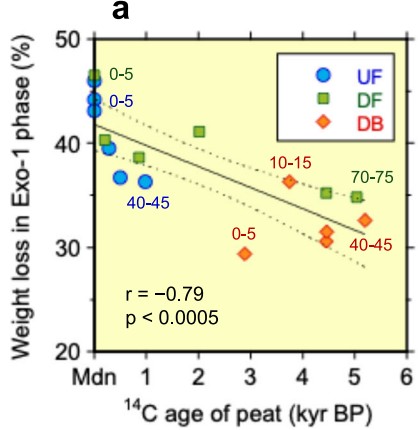

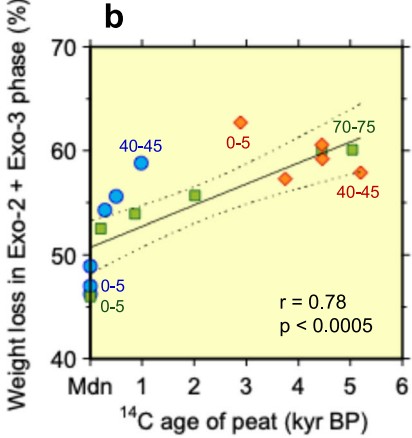

**Fig. 2 | Relationships between thermogravimetry weight loss and the $^{14}$C age of peat.** UF: undrained swamp forest, DF: drained forest, and DB: drained, burnt ex-forest. Mdn on the *X*-axis denotes the modern $^{14}$C age (set to 0 kyr BP for calculations). The weight loss associated with three temperature ranges (Exo-1, Exo-2, and Exo-3), each corresponding to the observed exothermic peak, was determined using simultaneous thermogravimetry and differential thermal analysis (Supplementary Figs. 2 and 3 and Supplementary Table 3). Weight loss in the Exo-1 phase (**a**) reflects the relative abundance of labile organic compounds (e.g.,

polysaccharides), whereas that in the Exo-2 + Exo-3 phase (**b**) reflects the relative abundance of recalcitrant, aromatic-rich organic compounds (e.g., lignin, humic substances)[29,30,50]. The 95% confidence intervals (upper and lower bounds) are indicated by the dashed lines. The peat depth (cm) is shown numerically. Linear relationships are observed; correlation coefficients ($r$) and $p$ values are provided in the figure. Individual weight losses in the Exo-2 and Exo-3 ranges did not correlate with $^{14}$C ages (Supplementary Fig. 4).

UF and DF sites (named PSF1 and PSF3, respectively) from June 2008 to May 2009 and estimated annual fluvial C discharges to be 63 and 88 g C m$^{-2}$ y$^{-1}$ for the PSF1 and PSF3 areas, respectively. Combined, these estimates suggest a drainage-induced peat-C loss of 1.8 kg C m$^{-2}$ (1.4 kg C m$^{-2}$ as $CO_2$ emissions and 0.5 kg C m$^{-2}$ as fluvial discharge) at the DF site over the 18-year period (1996–2014) following drainage. This value is significantly lower than the drainage-induced peat C loss of 5–11 kg C m$^{-2}$ that was evaluated on the basis of peat-C profile measurements in our study.

This discrepancy may be due to an underestimation of peat C loss from earlier flux measurements. $CO_2$ emissions through oxidative peat decomposition often peak in the early stages of drainage and subsequently decline over time[32,37], as readily decomposable organic compounds are preferentially consumed by soil microorganisms[37]. Given that drainage at the DF site began in 1996,

$CO_2$ emissions in the initial years were almost certainly higher than those measured in 2014. Furthermore, the unusually low GWL during the 2014 El Niño drought (mean GWL: −0.23 m)[20] compared with a mean GWL of −0.08 m in normal years (7 years during 2002–2018)[10] likely elevated $CO_2$ emissions, particularly at the undrained UF site. The elevated emissions at UF in 2014, likely due to drought conditions, reduce the observed difference between UF and DF, resulting in an underestimation of the drainage effect under normal conditions. Therefore, the drainage-induced $CO_2$ emissions over 18 years would be larger than the estimated value of 1.4 kg C m$^{-2}$. In addition, fluvial organic C release can increase immediately following drainage owing to increased surface runoff and water discharge[38,39], and the degassing of $CO_2$ from the water surface of peatland streams can also be significant[40,41], representing additional, unquantified pathways for peat-C loss.

Conversely, other studies indicate potentially greater peat-C losses from drained tropical peatlands than our estimate[42–44]. Couwenberg et al.[42] modelled a strong relationship between subsidence, as a measure of net peat-C loss, and GWL and reported an increase of 250 g C m$^{-2}$ y$^{-1}$ for every 0.1 m decrease in GWL. By applying this relationship to the difference in GWL between the UF and DF sites, we estimate an additional peat-C loss ranging from 14.4 to 18.5 kg C m$^{-2}$ over the 18 years of drainage. Similarly, Hirano et al.[43], using data from our DB site, reported that $CO_2$ emissions from oxidative peat decomposition increased by 89 g C m$^{-2}$ y$^{-1}$ for every 0.1 m decrease in GWL. This result suggests an additional peat-C loss of 5.1 to 6.6 kg C m$^{-2}$ over the 18-year period, which could reasonably explain our estimate.

Peat fires release substantial amounts of $CO_2$ into the atmosphere through peat combustion[12], yet the full extent of their impacts on C dynamics in tropical peatlands, including the ages of C released and postfire consequences, remains largely unknown[2,43,45]. Our results indicate that approximately 23–32 kg m$^{-2}$ of peat C was lost from the upper 0.4–0.5 m of the peat profile at the DB site due to repeated fires between 1997 and 2014 (Fig. 1, Supplementary Methods 2 and 3, and Supplementary Fig. 5). At this site, C emissions through peat combustion in 2002 were estimated to be 17.5 kg C m$^{-2}$ (ref. 45), whereas peat did not undergo combustion owing to fires in 2009 and 2014. In addition, cumulative C emissions through peat decomposition between 1997 and 2014 are estimated to be 6.8 kg C m$^{-2}$ from the GWL using a logarithmic equation[10]. Emissions from the 1997 fire are unknown, but the total of 24.3 kg C m$^{-2}$ is comparable to our result. This fire-induced peat C loss is more than 40 times the annual $CO_2$ emission through oxidative peat decomposition at the DB site[20] (Table 1). Crucially, the $^{14}$C analysis of the peat profiles reveals that these fire events released peat C accumulated over nearly 3000 years (Fig. 1) into the atmosphere, mirroring recent observations of millennium-aged peat burning in Indonesia on the basis of $^{14}$C measurements of fire-emitted particulate matter[46].

The combustion of the upper peat layer created a new surface, exposing previously protected deeper peat-C to oxidative decomposition. This deeper peat exhibits poorer substrate quality and contains a lower proportion of labile organic compounds than the upper peat does[37,47] (Fig. 2 and Supplementary Fig. 3). Peat burning can also modify the chemical structure of peat, often increasing its stability against decomposition through the production of pyrogenic aromatic compounds[48,49]. However, our thermal analysis reveals an abundance (more than 30%) of labile organic compounds, including polysaccharides (e.g., cellulose), in the peat at the surface of the DB site, with no signal of increasing highly recalcitrant compounds (e.g., lignin, aromatic structures)[30,50] (Fig. 2 and Supplementary Figs. 2 and 3). This finding suggests that the upper peat at the DB site remains highly susceptible to oxidative decomposition, even with its ancient age, although a significant relationship between the relative abundance of labile organic compounds and $^{14}$C age of peat was observed across all sites and depths (Fig. 2). Notably, significant $CO_2$ emissions, comparable to those observed at non-fire-affected sites, were still recorded at the fire-degraded DB site in 2014 (ref. 20) (Table 1). Collectively, these findings demonstrate that fire events initiate cascading oxidative decomposition of millennia-old peat, leading to significant postfire consequences. This result contrasts with the observations of Lupascu et al.[24], who reported modern $^{14}$C ages of $CO_2$ emitted from a degraded burnt tropical peatland in Brunei Darussalam; however, this discrepancy is likely due to the relatively young (approximately <100 years BP) upper peat layer at their burnt site. This suggests either a thick modern peat layer in the pre-fire profile, or the loss of only shallow surface peat layers resulting from seven fires that occurred between 1998 and 2016.

The radiocarbon ages of DOC in groundwater clearly differ among the three study sites (Fig. 3 and Supplementary Table 4), supporting the findings from the peat profile $^{14}$C analysis. The DOC-$^{14}$C ages at the

UF site indicate rapid turnover, originating from the oxidative decomposition of modern peat within the surface 0–20 cm layer (Fig. 1), which is above the mean GWL ($-0.15 \pm 0.13$ m, Table 1). In contrast, the DOC at the DF site is derived from the decomposition of centuries- to millennia-aged peat, mirroring the estimated $^{14}$C ages of peat-C lost from the peat profile during the 18 years of drainage. Moore et al.[8] previously confirmed these patterns, reporting similar $^{14}$C results for DOC samples collected at the outlets of channels draining the UF and DF sites in August 2008 (dry season) and May 2011 (wet season), identifying the UF DOC source as modern while revealing DF DOC-$^{14}$C ages ranging from 92 to 972 years BP. The presence of centuries-old DOC has also been documented in drainages from disturbed tropical peatlands in Sarawak[23] and Selangor[41], Malaysia.

Crucially, this study shows $^{14}$C ages of DOC across the disturbance sequence, which provides insights into the impact of peat fires on DOC source and dynamics, extending beyond previous reports that described changes in the concentration and quality of DOC due to drainage and fires[24,49,51,52]. Radiocarbon data from groundwater at the repeatedly burnt (DB) site reveal that DOC originates from the oxidative decomposition of peat preserved for up to 4000 years, which is significantly older than the DOC found at the undrained (UF) and drained (DF) sites (Fig. 3) and aligns with the peat-C ages at the DB site (Fig. 1a). These findings provide compelling evidence for the fire-induced release of ancient peat-C from tropical peatlands, highlighting the profound vulnerability of this globally important, long-term C sink to anthropogenic disturbances. Notably, a comparably old $^{14}$C age has been reported for DOC from channels draining oil palm plantations in peninsular Malaysia, representing the oldest recorded age for soil-derived natural surface-water DOC[8].

Tropical peatlands play a vital role in climate change mitigation, as they have sequestered 105 Pg of C within their belowground peat profiles in the Holocene[2,6]. However, our results show that peat C is highly vulnerable to loss under human-induced disturbances such as drainage and fire, indicating that its long-term stability cannot be guaranteed under current land-use practices. The pronounced shifts in C stock and release processes triggered by disturbances are shown in Fig. 4, revealing a clear pattern of accelerated loss of ancient peat C. Recovery of this lost C requires millennia[53]. Reclaiming drainage channels is considered a strategy to reduce C emissions and recover peat C stock[54]. However, fully restoring the original forest cover, particularly in burnt areas, is challenging due to a lack of seeds and nutrients[54], and postfire flooding caused by land subsidence from peat loss[2] (Table 1). Projected increases in temperature and GWL in this region[55] further complicate the outcome of peatland C recovery under a changing climate.

Using $^{14}$C analysis, we quantify substantial C loss from both our drained and repeatedly burnt sites and report that drainage and subsequent peat fires released approximately 30–41 kg C m$^{-2}$ over 18 years, which is sufficient to transform the peat swamp forest from a C sink to a significant source of emissions[9,10,28]. By extrapolating our results to drained and fire-damaged peatland areas of Indonesia (Supplementary Table 5), we estimate that an additional 0.81–3.70 Gt of peat carbon could have been released due to drainage (0.48–1.49 Gt C) and repeated fires (0.33–2.21 Gt C) between 1996 and 2014 (Supplementary Table 5). The C emissions from fires align with estimates (1.3 Gt C over 18 years) for the Equatorial Asia region, based on the Global Fire Emissions Database[56]. Our estimated release from drained and burnt areas represents approximately 1.4–6.4% of Indonesia's stored peatland C (57.4 Gt C)[1]. Critically, in addition to the immediate C emissions from fires, ongoing decomposition of ancient peat in drained peatlands (10.9–13.5 Mha) releases an additional 0.03–0.08 Gt C annually—approximately 1.7–5.2% of the global land net C sink (currently estimated at 1.6 Gt C y$^{-1}$)[57]. Our approach allows for the evaluation of C loss and quality, accounting for the integrated effects of disturbances, regardless of peat subsidence and C loss pathways. A

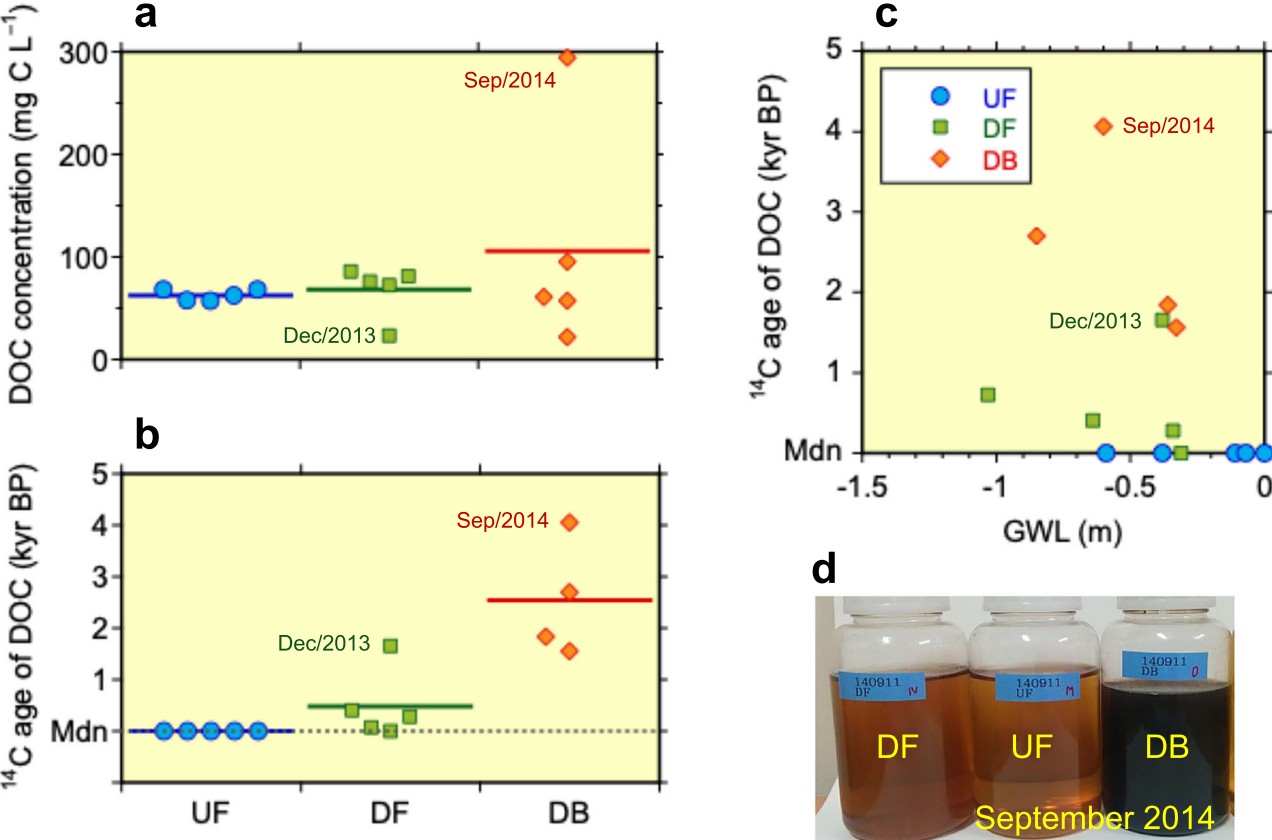

**Fig. 3 | Dissolved organic carbon (DOC) characteristics.** Concentrations (**a**), DOC-[14]C ages (**b**), and the relationships between the DOC-[14]C age and groundwater level (GWL) (**c**) in groundwater samples (**d**) collected from 2013 to 2016 at three peatland sites in Palangka Raya, Indonesia. UF undrained swamp forest, DF drained forest, and DB: drained, burnt ex-forest. Mdn on the *Y*-axis denotes the modern [14]C age. The Sep/2014 sample from the DB site was collected shortly after a fire event. The GWL data are from Hirano et al.[10].

key limitation of this study is the lack of replicated peat profile samples, which restricts a comprehensive assessment of heterogeneity and uncertainty, particularly at the burnt site. Future research will expand upon this approach in the same and other peatland ecosystems to provide a robust assessment of the regional impacts of peatland disturbances on the global C cycle.

## Methods

### Study site

This study was conducted at three peatland sites with varying degrees of degradation: an undrained swamp forest (UF), a drained forest (DF), and a drained, repeatedly burnt ex-forest with limited vegetation (DB). These sites are located within 15 km of tropical peatlands near Palangka Raya, Central Kalimantan, Indonesia, and are situated within Block C of the former Mega Rice Project[12]. These sites have been used to investigate the effects of disturbances on $CO_2$ emissions from peat decomposition and the C balance in tropical peatlands[8,10,12,20,45,55,58]. Detailed descriptions of the sites can be found in Hirano et al.[58], Supplementary Method 1 and Supplementary Table 2.

These sites are situated relatively close to the edge of peat domes and have similar topographies, vegetation types, and peat thicknesses (Table 1 and Supplementary Table 2)[8,20]. All sites previously underwent selective logging until the late 1990s. The UF site is a swamp forest dominated by *Combretocarpus rotundatus* and *Cratoxylum arborescens*, with a rich understory of shrubs largely composed of their sapling[58–60]. The site was in a National Park designated in 2006 and was slightly drained by small ditches made for logging; however, the shallow ditches were mostly buried naturally[10]. The DF and DB sites, formerly undrained swamp forest, have been drained by canal

excavation since 1996 (ref. 54). The DB site experienced repeated fires in 1997, 2002, 2009, and 2014, i.e., El Niño years, resulting in the repeated loss of surface peat and vegetation (shrubs, ferns, and grasses). The mean annual temperature and precipitation from 2002–2018 were 26.2 ± 0.27 °C and 2557 ± 432 mm, respectively[10]. The key site characteristics, including the ecosystem-scale C balance, $CO_2$ emissions from peat decomposition, and groundwater level (GWL), are summarized in Table 1.

### Collection and analysis of peat samples

Peat sampling was conducted via the pit excavation method at all three sites to prevent contamination of samples in the depth direction. Sampling was performed in September 2014, considering the generally lowest GWLs of the year and the unusually low GWLs associated with the 2014 El Niño drought[20]. Briefly, a soil pit was dug at least 1 m apart from any trees at each site, and peat samples were collected from selected layers at 5-cm depth intervals using a 100-cm³ stainless steel cylinder. The maximum sampling depth varied among the sites (45 cm, 75 cm, and 45 cm for the UF, DF, and DB sites, respectively), depending on the GWL. Although the sample size was small (17 for three profiles), the results indicate well-preserved, stratified profiles from the surface to the bottom for each site. The sampling did not provide spatial coverage for each site.

In the laboratory, peat samples were oven-dried for 96 h at 105 °C and ground in a mortar. The dry bulk density was calculated for each 5-cm layer by dividing the dry weight of the sample by its volume (100 cm³). The carbon content of the samples was determined using a CN analyser (Sumigraph NCH-22, Sumika Chemical Analysis Service Ltd., Japan), as previously reported by Itoh et al.[20]

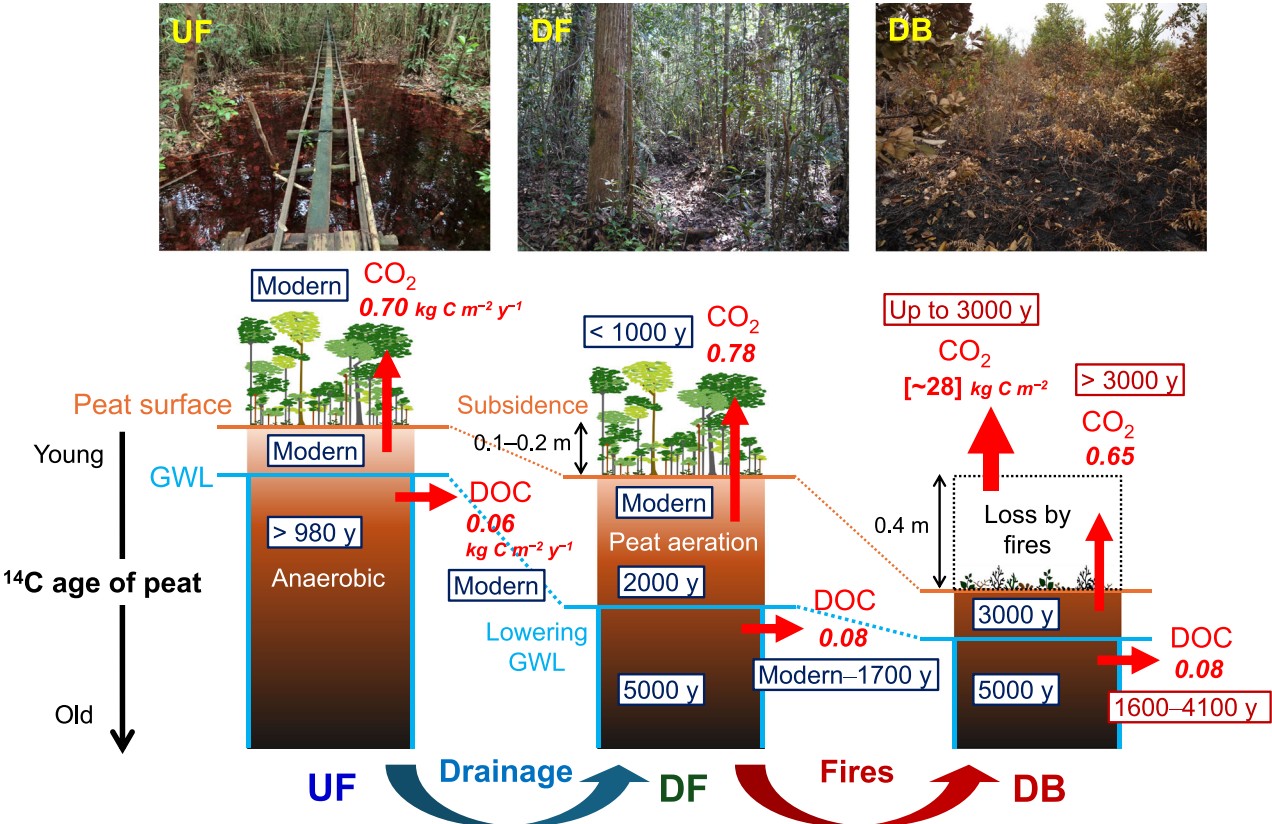

**Fig. 4 | Impact of drainage and subsequent fires on carbon stock and release in the tropical peatland in Palangka Raya, Indonesia.** Sites represent different disturbance histories. UF: undrained swamp forest, DF: drained forest, and DB: drained, burnt ex-forest. Conventional ¹⁴C ages (in years BP) of stored and released C are shown in boxes, and C fluxes (in kg C m⁻² y⁻¹) are italicized. Flux data are from Itoh et al.[20] for $CO_2$ and Moore et al.[8] for dissolved organic C (DOC).

Radiocarbon analysis was conducted on subsamples (equivalent to approximately 4 mg of C) following the methods of Koarashi et al[61]. The radiocarbon content of the samples was measured using an accelerator mass spectrometer (JAEA–AMS–MUTSU) at the Japan Atomic Energy Agency (JAEA). The dried samples were combusted with clean CuO wire and Ag foil in evacuated and sealed quartz tubes for 2 h at 850 °C. The resulting $CO_2$ was cryogenically purified and converted to graphite on iron powder by reduction with $H_2$ gas at 650 °C, producing graphite targets for accelerator mass spectrometry (AMS)-¹⁴C analysis. In this study, the ¹⁴C results were normalized to a common $\delta^{13}$C value of −25‰ and reported as the conventional ¹⁴C age (years before present, BP)[62], with an analytical uncertainty of less than ± 60 years (at one standard deviation). Calibrated ages of peat, calculated using Calib 8.20 software[63] with the IntCal20 radiocarbon calibration curve[64], are also provided in Supplementary Table 6.

The thermal decomposition behaviour of the peat samples was investigated using simultaneous thermogravimetry (TG) and differential thermal analysis (DTA). Analyses were performed using either a ThermoMass Photo or a TG-DTA8122 (both from Rigaku, Japan). Approximately 10 mg of each sample was heated in an alumina crucible from room temperature to 650 °C under a constant airflow rate of 300 ml min⁻¹ and a heating rate of 3 °C min⁻¹. To differentiate labile and recalcitrant organic compound groups with different structural compositions, a 90-min isothermal hold at 375 °C was used[30,50]. The DTA curves of the peat samples generally exhibited three exothermic peaks (Supplementary Method 4 and Supplementary Fig. 6), although the profile of the second peak was affected by the isothermal hold. TG data were then used to quantify weight loss across three defined temperature ranges (Exo-1, Exo-2, and Exo-3), each corresponding to the observed exothermic peak (Supplementary Table 3).

### Collection and analysis of water samples

Groundwater sampling was conducted at all three sites on five occasions between 2013 and 2016: December 2013, September 2014, December 2015, June 2016 and August 2016. Groundwater samples were collected from near the groundwater surface using wells with strainers (with a maximum depth of 1.8 m below the peat surface). Immediately after sampling, the water samples were filtered through prebaked (for 4 h at 450 °C) GF/F filters (Whatman) and stored in prerinsed polycarbonate bottles at 4 °C.

In the laboratory, the DOC concentrations of the samples were determined using a total organic carbon analyser (TOC-V, Shimadzu, Japan). The groundwater samples were subsequently stored at 4 °C under low pH conditions until freeze-drying. In October 2020, DOC samples were obtained as powdered samples by freeze-drying approximately 60–250 ml of the stored samples after confirming no changes in DOC concentration during storage. For two samples, an aliquot was taken immediately after the DOC concentration was measured, and DOC samples were obtained by freeze-drying and stored as powdered samples in sealed containers until 2020. These two samples were also used for confirmation of storage effect. AMS-¹⁴C analysis was performed on subsamples (equivalent to approximately 1 mg of C) from both sets of DOC samples.

The powdered DOC samples were ground and prepared for AMS-¹⁴C analysis using the same procedure as that used for the peat samples. The radiocarbon content was measured using an accelerator mass spectrometer (JAEA–AMS–TONO–5MV) at the JAEA. The ¹⁴C data were reported as conventional ¹⁴C ages (years BP). Calibrated ages of DOC are also provided in Supplementary Table 7.

## Quantifying peat carbon stocks

The peat C stock (kg C m$^{-2}$) within a 5-cm layer was calculated by multiplying the C content (kg C kg$^{-1}$) of the peat sample by the bulk density (kg m$^{-3}$) and the thickness (5 cm) of the layer. Owing to incomplete sampling of the entire peat profile, peat samples were collected from selected layers at 5 cm depth intervals (Supplementary Table 1). To estimate the cumulative peat C stock from the surface to the bottom sampled layer, the C stock in the uncollected layers was estimated as the mean value of the C stocks immediately above and below the target layer (Supplementary Table 1).

## Evaluating C losses due to drainage and fires

Peat-C losses due to drainage and fires were evaluated using cumulative C stocks (kg C m$^{-2}$) as a function of $^{14}$C age (kyr BP) for the three sites. The drainage-induced loss of C accumulated over the past 1,000 years was estimated as the difference in cumulative C stock between the DF and UF sites (UF minus DF) at a $^{14}$C age of 1 kyr BP (Fig. 1b, Supplementary Fig. 1, and Supplementary Method 2). Similarly, the fire-induced loss of C was estimated as the difference in cumulative C stock between the DF and DB sites (DF minus DB) at a $^{14}$C age of 2.89 kyr BP, corresponding to the $^{14}$C age of peat in the uppermost layer of the DB site (Fig. 1b, Supplementary Table 1, and Supplementary Method 2).

## Extrapolation to Indonesia's peatland

Our estimates of C losses due to drainage and fires were extrapolated to the entire area of Indonesia, considering both drained and fire-damaged peatland areas (Supplementary Table 5). The area of drained peatland was estimated by multiplying the total peatland area (20.70 Mha)[1] by an assumed proportion of drained peatland (65% or 48%). The 65% proportion is an estimate for an area encompassing 157,000 km$^2$ across Sumatra, Borneo, and Peninsular Malaysia, derived from a regional map of drainage canals[31]. The 48% proportion is based on decadal mean peatland area data for different land cover types from 2011 to 2020 (ref. 55). The proportion was calculated by summing the areas of drained peat swamp forest, oil palm plantation, farmland, and drained shrubland, and dividing the sum by the total peatland area in Indonesia (excluding Papua), assuming land-use change from previously undisturbed peat swamp forest. The average groundwater level difference between UF and DF is 0.32 m (ref. 10), which is nearly the same as the 0.33 m average groundwater level difference between undrained and drained forests in the study area (Sumatra, Borneo, and the peninsula Malaysia) over a 10-year period[55]. Therefore, the drainage intensity of DF can be considered approximately equivalent to that of drained forests in the tropical peatlands of Southeast Asia. The area of fire-damaged peatland was based on estimates by Page et al.[12], representing an intermediate estimate of 2.44 Mha, with lower and upper bounds of 1.45 and 6.80 Mha, respectively. Note that this extrapolation assumes that drained and fire-damaged areas in Indonesia exhibited similar disturbance levels to our sites during the period 1996–2014.

## Reporting summary

Further information on research design is available in the Nature Portfolio Reporting Summary linked to this article.

## Data availability

The data generated in this study are provided in the Supplementary Information file.

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

## Acknowledgements

We thank Kikuko Yoshigaki, Mutsumi Shimizu, and Misuzu Kaminaga of the JAEA for their assistance with the laboratory work in this study. This work was supported by JSPS KAKENHI (Grant Number 19H05666 to T.H.) and the Research Institute for Humanity and Nature (RIHN; Project No. 14200117 to M.I.).

## Author contributions

J.K., M.I. and T.H. conceived the study. M.I., K.K., T.H., A.J. and S.D. collected the samples. M.I. and M.A.A. conducted the elemental analysis. J.K., M.A.A. and Y.S.K. conducted the radiocarbon analysis. M.A.A. and M.M. conducted the thermal analysis. J.K. conceived the paper and wrote the initial draft, to which all authors provided critical contributions and approved submission.

## Competing interests

The authors declare no competing interests.
