## [Transparent Peer Review file · Nature Communications]

Progressive release of long-stored carbon from tropical peatland disturbances

Corresponding Author: Dr Jun Koarashi

Version 0:

Reviewer comments:

Reviewer #1

(Remarks to the Author)

Reviewer #2

(Remarks to the Author)

Line 44 – Any justification for using the term PSF vs. peatland? The paper starts by using peatland, then switches to PSF, and later returns to peatland again. Preferably, just use the term peatland consistently.

Line 66 – Please revise the sentence to specify that UF, DF, and DB are peatland sites where the samples were collected. Currently, it sounds like they are three types of PSFs.

Line 385 – Is there any radiocarbon date calibration? It needs to be specified in the methods section: what software was used to calibrate the dates, which calibration curve was used, etc. It is also preferable to calibrate the radiocarbon data and present the ages in cal BP instead of kyr BP (uncalibrated).

Fig. 1a – Instead of using linear age, can you conduct an age–depth analysis (e.g., Bacon) and integrate the results into this graph?

Figs. 1 and 2 – Can you please specify on the x-axis that the radiocarbon dates are from peat soil samples (maybe “¹⁴C age of peat soil”)? You have two sets of dates—one for peat soil and one for DOC. In Fig. 3 it is clearly labeled as DOC dates (“¹⁴C age of DOC” on the x-axis), but in Figs. 1 and 2 it needs to be specified more clearly.

Fig. 4 – Great figure, very informative. Can you also add the amount of subsidence from UF to DF? Since you mention that subsidence from DF to DB is 40 cm, it would be better if you also mention the value for UF to DF.

Reviewer #3

(Remarks to the Author)

General comments

The proposed study investigates the impact of drainage induced aerobic decomposition and fires on carbon gas emissions from tropical peatlands in Indonesia, using the radiocarbon (¹⁴C) method. Southeast Asian tropical peatlands represent an exceptionally large but highly threatened carbon storage, which is rapidly losing due to drainage and fires. This research aims to assess changes in peat quantity through an approach that is less dependent on measurement and prediction errors, as well as sample representativeness, issues that often affect conventional soil carbon flux measurement techniques such as so called chamber methods.

Radiocarbon analysis is a relevant tool for determining peat age, particularly given that the age range of peat layers falls

within the time range where the method is most reliable. The applied approach, comparing age estimates across sites subjected to different treatments (undrained control, drained, and burned), is valid in that it avoids biases in the estimates of mass loss caused by e.g. peat subsidence. However, age observations from the undrained site do not extend very deep in the peat layer, and it would have been interesting to see how the ages of the deeper peat layers are. There is strong reason to assume a linear correlation between peat age and depth (Fig. 1), but this relationship varies among sites; it is clearly steepest at the undrained site. Further clarification of these differences would be valuable; do they result from insufficient observations or from “genuine” site-specific variations? This question is important, as it influences quantitative estimates of carbon losses.

An important question is whether the studied sites can truly be considered representative of one another in terms of peat layer characteristics. If this assumption can be credibly justified, the knowledge produced by this study would have significant global relevance and could fundamentally reshape estimates of carbon loss from tropical peatlands, associated risk assessments, and projections of future carbon emissions. At least, the study is a promising step forward in the development of GHG inventory methodologies and introduces a novel assessment approach that could be applied to peatlands worldwide. I recommend acceptance of the manuscript following the above-mentioned revisions.

Detailed comments

r.133: This is quite a drastic result, but the order of magnitude seems to be correct.

r.146: It is noteworthy that the exposure of deeper layers to aerobic decomposition did not differ from the decomposition potential of surface peat layers based on chemical characteristics.

r.187–189: Here, the emissions from drainage and fires appear to have been extrapolated to the entire peatland area of Indonesia (20.7 M ha)? However, it should be noted that this area includes all peatlands, including those in natural undrained state. The calculation therefore seems overestimated if the assumption is that this entire area has burned and been drained?

Table 1: It is remarkable that respiration is not highest at burned and drained sites but at the undrained site, where high GPP compensates for losses. Could a note be added here on the method used to measure respiration?

r.378: How was the starting level (0-level) for peat sampling defined at the different sites?

r.409–411: The number of water samples taken is very low. It is difficult to assess intra-annual variation in concentrations based on these sampling frequency, especially after fire. Have the DOC concentrations been measured in previous studies with higher resolution that could be cited?

Version 1:

Reviewer comments:

Reviewer #2

(Remarks to the Author)

The author(s) have adequately addressed all reviewer comments, therefore, I recommend acceptance.

Reviewer #3

(Remarks to the Author)

I have carefully reviewed the revised manuscript and read through the response letter, including the authors' responses to my own comments and those of the other reviewers. In my opinion, the authors have taken into consideration all the aspects of the study, which were under my criticism and significantly improved the manuscript, particularly concerning the methods and reliability of the results. I have also examined the statistical methods and the statistical results presented in the body and supplementary material of the manuscript. Based on this assessment, I support the acceptance of the manuscript.

Cascading release of ancient carbon from tropical peatland disturbances

Jun Koarashi et al.

General Comments and Main Concerns

The manuscript addresses an important and timely topic: carbon (C) dynamics in tropical peatlands across disturbance gradients, from undrained peat swamp forests to drained and fire-impacted areas, using ^{14}C dating and dissolved organic carbon (DOC) analysis. While the research question and most of the methods are relevant, the current version of the manuscript has significant shortcomings that undermine its conclusions.

The study has potential to contribute valuable insights into C dynamics in tropical peatlands, but the current manuscript is not acceptable in its present form. Correcting the noted shortcomings is essential to enhance the validity and credibility of the study. A major revision is required.

Major concerns include:

1. **Oversimplified assumptions and limited data**

The study relies on overly simplified assumptions about peatland characteristics. The sample size is very small (only three profiles and 15 samples), with no spatial replication or coverage across different peatland types. This makes it highly questionable to generalize findings to all Indonesian peatlands.

2. **Lack of uncertainty analysis and sensitivity testing**

Extrapolations are presented without confidence intervals, error ranges, or assumptions. No sensitivity analysis is provided, despite many values being estimated rather than measured (e.g., peat dry bulk density, CO_2 emissions, fire-related C losses). This omission severely limits the reliability of the conclusions.

3. **Insufficient site characterization and methodological clarity**

Critical details about study sites such as peat thickness, drainage intensity, ditch spacing, vegetation composition are missing or vague. Sampling strategy and collection is unclear (e.g., depth selection, number of samples per site, rationale for layer selection). These gaps make it difficult to assess the validity and comparability of the data.

4. **Literature integration and discussion**

The discussion does not adequately engage with existing studies on tropical peatland C dynamics, drainage and fire impacts. Several relevant references are missing, and the manuscript underestimates prior research on DOC and C loss processes.

5. **Terminology and consistency issues**

Key terms (e.g., 'disturbance gradients,' 'release,' 'immense C reservoir') are vague or ambiguous. Acronyms are inconsistently used, and units are not standardized throughout the text. Expressions like "vast quantities" or "unsustainable" need to be replaced with precise, quantitative language.

6. **Extrapolation and interpretation**

Statements about C release (e.g., 1.38 Gt C) lack clarity on whether they refer to gross emissions, net flux, or cumulative losses. The basis for extrapolation is unclear, and

the manuscript does not clearly distinguish between undrained and drained peatlands in its analysis, nor are the expressions sufficiently clear.

7. Writing quality and clarity

Numerous sentences are unclear, overly complex, or logically inconsistent. The abstract and key paragraphs need restructuring for readability and scientific precision.

Specific Comments

Line 1: Consider a more informative topic, such as 'Progressive release of long-stored carbon due to disturbances in tropical peatlands'

Lines 23–27: The sentence is unclear. Consider rephrasing for clarity, for example: *'During 18 years, drainage and subsequent peat fires released approximately 36 kg C m⁻² yr⁻¹, of which drainage accounted for 8 kg C and fire events for approximately 28 kg C.'*

Lines 27–29: The phrase *'disturbance gradients'* is vague. Specify the types of disturbance (e.g., logging, drainage, fire). The reference to *'from the present to more than 4000 yrs BP'* suggests a temporal range, but the sentence implies a spatial gradient linked to disturbance. Clarify whether you mean spatial variation or time progression. Suggest stating that disturbances mobilize older peat C.

Line 29: Clarify here what is meant by *'Indonesia's peatlands'*. Do you refer to drained peatlands, peat swamp forests (PSFs), or all peatlands including swamps? Note that not all swamps are peatlands. This distinction is important for interpreting potential C release. I notice that some of the information is provided in 'Extended Data Table 4,' but essential and important details should also be included in the main manuscript so that the context and magnitude are clear to the reader.

Lines 29–32: The phrase *'a potential release of 1.38 Gt C between 1996 and 2014'* is ambiguous. Does this refer to net emissions (emissions minus sequestration), gross emissions, or cumulative losses? Specify the source of release (peat soils?) and the basis for extrapolation (flux measurements, land-use change data, or model outputs). Include uncertainty estimates—confidence intervals or assumptions are standard for extrapolations.

Line 36: The term *'immense C reservoir'* is vague. Use quantitative language or comparisons, e.g., *'a reservoir containing X Gt of C.'*

Line 40: Rephrase *'harbours the largest swath of these ecosystems'* to something clearer, such as *'contains the largest area of these ecosystems.'*

Line 47: Specify the total area when stating *'less than 40% of native PSFs remaining undisturbed.'*

Lines 54–55: Clarify *'28-fold greater than predisturbance rates of loss.'* Greater than what baseline? Provide context.

Lines 55–57: Clarify whether *'mobilized from deeper peat layers'* refers to C loss from drained and burned areas.

Line 57–59: The phrase *'previously unexplored C pool'* is unclear. If you mean C loss from deeper peat layers (CO₂, DOC) not fully considered before, state that explicitly.

Lines 60–63: The claim that comprehensive assessments are lacking seems overstated. Several studies have examined drainage and wildfire impacts on tropical peatlands. Revise and cite relevant literature. Some examples are given below (see comments for lines 167–176).

Lines 66–67: Avoid nested acronyms. Use simpler acronyms like UPSF and DPSF for undrained and drained PSF consistently.

Line 67: Clarify “*edaphically similar area*” in relation to your research sites.

Line 69: Use ‘*peat thickness*’ instead of ‘*peat depth*’ throughout the manuscript.

Line 74: Avoid vague phrases like ‘*this vital C reservoir.*’ Use precise language.

Lines 72–85: The paragraph is unclear and difficult to follow, similar to the abstract.

Restructure for clarity, including total C loss and its components. For example:

‘During 18 years, drainage and subsequent peat fires released approximately 36 kg C m⁻² yr⁻¹, of which drainage accounted for 8 kg C and fire events for approximately 28 kg C.’

Lines 83–85: The statement on CO₂ emissions aligns with previous GHG balance studies of drained peatlands. Acknowledge prior research and cite references.

Lines 107–113: There is a logical conflict with lines 95–99. The text should clarify that elevated emissions at UF in 2014 (likely due to drought) reduce the observed difference between UF and DF, leading to an underestimation of the drainage effect. The measured difference (6 kg C m⁻² yr⁻¹) is smaller than it would be under normal conditions; therefore, the true drainage effect (the difference between undrained and drained sites under typical conditions) is larger than 6 kg C m⁻² yr⁻¹ over 18 years.

Line 114: Clarify ‘*potentially greater peat-C losses.*’ Greater than what?

Line 124–125: Explain what is meant by ‘*full extent of their impacts on C dynamics remains largely unknown.*’ Beyond CO₂ emissions, what aspects are unknown?

Line 125 and throughout: Ensure consistency in units (e.g., use m⁻² instead of “per square meter”).

Lines 125–126: Clarify. I guess you mean simply that 28 kg C m⁻² was lost from the upper 0.4 m of the DPSF.

Line 135: Rephrase ‘*fire events liberated peat C*’ to ‘*fire events released peat C to the atmosphere.*’

Lines 143–147: This part is hard to follow and has too many nested ideas. Thermal analysis → abundance of labile compounds → example (cellulose) → no signal of flame-retardant materials → susceptibility → ancient age. Simplify and clarify. Replace “*flame-retardant organic materials*” with ‘*highly recalcitrant compounds (e.g., lignin, aromatic structures).*’ Explain the link between age and susceptibility more clearly.

Lines 151–154: Clarify whether Lupascu et al.’s sites differ from yours. Provide context for comparing surface layers.

Line 159: State the mean GWL explicitly and reference Table 1.

Lines 167–176: Revise this paragraph. Prior studies have examined drainage and fire effects on DOC. Cite relevant references and compare findings in the discussion. Some examples:

Yang, P. et al. (2025). Heating-Induced Redox Property Dynamics of Peat Soil Dissolved Organic Matter in a Simulated Peat Fire: Electron Exchange Capacity and Molecular Characteristics. *Environmental Science & Technology*, 59, 489–500.

Sundari, S. (2020). “Dissolved organic carbon and physicochemical variables of peat water in tropical peat swamp forests.” IOP Conference Series: Earth and Environmental Science, 591, 012045.

Sazawa, K., Wakimoto, T., Fukushima, M., Yustiawati, Y., Syawal, M. S., Hata, N., Taguchi, S., Tanaka, S., Tanaka, D., & Kuramitz, H. (2018). Impact of peat fire on the soil and the export of dissolved organic carbon in tropical peat soil, Central Kalimantan, Indonesia. *ACS Earth and Space Chemistry*, 2(7), 692–701. <https://doi.org/10.1021/acsearthspacechem.8b00018>

Lupascu, M., Akhtar, H., Smith, T. E. L., Sukri, R. S. (2020). Post-fire carbon dynamics in the tropical peat swamp forests of Brunei reveal long-term elevated CH₄ flux. *Global Change Biology*, 26(9), 5125–5145.

Burd, K., Estop-Aragonés, C., Tank, S. E., Olefeldt, D. (2020). Lability of dissolved organic carbon from boreal peatlands: interactions between permafrost thaw, wildfire, and season. *Canadian Journal of Soil Science*, 100(4), 503–515.

Sundari, S. (2020). “Dissolved organic carbon and physicochemical variables of peat water in tropical peat swamp forests.” IOP Conference Series: Earth and Environmental Science, 591, 012045.

Line 169: Clarify '*fire-affected (DB) site*.' Is this the repeatedly burnt drained site? Be consistent.

Line 171: Clarify '*severely drained (DF) sites*.' Maintain consistent terminology.

Line 178: Avoid vague phrases like '*sequestered vast quantities of C*.' Provide numbers.

Lines 178–183: 'readily reversed'. Does this mean rapid C loss or degradation, or both or something else?). Replace vague language with precise statements. You could say for example: '*Our results show that peat C is highly vulnerable to loss under human-induced disturbances such as drainage and fire, indicating that its long-term stability cannot be guaranteed under current land-use practices.*'

Line 180: Throughout the ms, ensure consistency in terms like '*C storage*,' '*stocks*,' and '*store*.'

Lines 182–183: Clarify whether recovery is likely/unlikely under changing climate conditions.

Line 184: Specify whether '*from our PSF*' refers to drained sites, repeatedly burnt sites or both.

Lines 187–189: Clarify whether extrapolation applies to all peatlands. Separate undrained and drained peatlands in analysis. See my earlier comment about Indonesia's peatlands.

Line 192: Provide the area of drained peatlands.

Lines 195–199: A sensitivity analysis is essential due to high uncertainty and small sample size. Use error ranges (e.g. NEE-values of Table 1) and compare bulk density values with other studies. Include references for dry bulk density and C concentrations.

References: Ensure uniform reference style.

Figure 1: Add GWL range for clarity.

Lines 356–362: Provide details on ditching and spacing at study sites. How are the sites comparable to drained sites of Indonesia? Define '*severely drained forest (DF)*.'

Line 362: Include key site characteristics (e.g. size, peat thickness) in Table 1 for context. At the moment, there is very little info of the sites, only a single reference to Hirano et. al.

Lines 365–367: *Please, be more specific.* Specify dominant native tree species'

Lines 367–368: Clarify '*a national park since at least 2006.*'

Line 366: Replace vague '*rich shrubbery understory*' with species names.

Lines 367–368: 'formerly a PSF'? Aren't all sites PSFs with different disturbance histories?

Lines 378–381: Lines 378–381: '*Selected layers*'. Explain how layers were selected (visual characteristics, stratigraphy, C content, something else). State sampling depth range and number of samples per site. Site-specific variation: Were the same depths sampled at all sites? Does the sampling begin at the surface and go down to a specific depth or only certain horizons were targeted? Note that the sample size (three profiles, 15 samples) is very small and lacks spatial coverage, limiting generalization.

Line 383: 'dry bulk density.'

Line 389: Specify whether subsamples or whole samples were used. Sample size?

Lines 401–402: Clarify whether TG-DTA was used to identify labile and recalcitrant compounds as two groups or within each category.

Line 417: State how long samples were stored at 4°C before analysis.

Lines 418–419: Clarify what is meant by '*some samples*.' How many? Were all samples analyzed?

Lines 430–431: Yes, incomplete sampling and small sample size are major limitations. This issue underlines the need for sensitivity analysis.

RESPONSE TO REVIEWERS

Reviewer #1 (Remarks to the Author):

General Comments and Main Concerns

The manuscript addresses an important and timely topic: carbon (C) dynamics in tropical peatlands across disturbance gradients, from undrained peat swamp forests to drained and fire impacted areas, using ^{14}C dating and dissolved organic carbon (DOC) analysis. While the research question and most of the methods are relevant, the current version of the manuscript has significant shortcomings that undermine its conclusions. The study has potential to contribute valuable insights into C dynamics in tropical peatlands, but the current manuscript is not acceptable in its present form. Correcting the noted shortcomings is essential to enhance the validity and credibility of the study. A major revision is required.

Response: We appreciate the constructive comments and detailed suggestions, which have provided valuable guidance for improving the manuscript. We have addressed each of your points below with specific revisions made to the text.

Major concerns include:

1. Oversimplified assumptions and limited data

The study relies on overly simplified assumptions about peatland characteristics. The sample size is very small (only three profiles and 15 samples), with no spatial replication or coverage across different peatland types. This makes it highly questionable to generalize findings to all Indonesian peatlands.

Response: According to this and other comments, we now present the ranges of estimates based on sensitivity analysis, including both our results and published information (such as drained and fire-damaged areas). Please see the response to major comment 2 below.

One of the strongest points of this research is that it was conducted at sites where extensive investigations have been conducted for a variety of ecological properties and C dynamic processes over several decades. This provides an invaluable opportunity for a robust validation of our findings obtained through an innovative approach, from various aspects including CO_2 release through oxidative peat decomposition, DOC

export via channels, drainage impact estimates based on groundwater levels, and fire impact estimates based on ecosystem C exchange. Although we acknowledge the limited sample size and lack of spatial replication, comprehensively assessing this existing knowledge allows for a thorough discussion and assessment of our results.

Regarding drained conditions at our sites, the average groundwater level difference between the undrained (UF) and drained (DF) sites is evaluated to be similar to the average groundwater level difference in the study area (Sumatra, Borneo, and the peninsula Malaysia) (see the response to specific comment 41 for details). Therefore, the drainage intensity of our drained sites can be considered similar to that of drained forests in Southeast Asian tropical peatlands. This supports our extrapolation estimates.

Regarding fire impacts, it would be difficult to assess the robustness of the extrapolation because severity can vary even within a site, and the frequency of fires can vary across Indonesia. However, our estimated 0.33–2.21 Gt C release from burnt areas of Indonesia aligns with estimates (1.3 Gt C over 18 years) for the Equatorial Asia region, based on the Global Fire Emissions Database (van der Werf et al., 2017). This point is now addressed in the last paragraph of the main text.

Currently, to our knowledge, no published data exists on spatial variability in ^{14}C age profiles in tropical peatlands. Clearly, our research emphasizes the need for future research with larger and more spatially diverse datasets, including within-site and between-site variability for a sequence of disturbances. As stated in the last sentence in the main text, this paper contributes to accelerating research by using the ^{14}C approach to provide an improved understanding of peatland C dynamics and a robust assessment of the regional impacts of peatland disturbances on the global C cycle.

2. Lack of uncertainty analysis and sensitivity testing

Extrapolations are presented without confidence intervals, error ranges, or assumptions. No sensitivity analysis is provided, despite many values being estimated rather than measured (e.g., peat dry bulk density, CO₂ emissions, fire-related C losses). This omission severely limits the reliability of the conclusions.

Response: Carbon losses due to disturbances have been evaluated based on differences in cumulative C stocks (kg C m^{-2}) as a function of ^{14}C age (kyr BP) between the two sites of interest (UF and DF for drainage-induced C loss and DF and DB for fire-

induced C loss). For all sites, the cumulative C stocks exhibited significant linear relationships with ^{14}C age (Fig. 1b; UF: $r = 0.99$, $p < 0.05$, DF: $r = 0.99$, $p < 0.0005$, DB: $r = 0.97$; $p < 0.01$) and were estimated as a function of ^{14}C age, with standard errors. Using these relationships and standard errors, the drainage-induced loss of C accumulated over the past 1,000 years was estimated as 8.0 kg C m^{-2} (range: $4.9\text{--}11.1 \text{ kg C m}^{-2}$) as the difference in cumulative C stocks between DF and UF (UF minus DF) at a ^{14}C age of 1 kyr BP. Similarly, the fire-induced C loss was estimated as 27.6 kg C m^{-2} (range: $22.8\text{--}32.5 \text{ kg C m}^{-2}$) as the difference in cumulative C stocks between DF and DB (DF minus DB) at a ^{14}C age of 2.89 kyr BP, the ^{14}C age of peat in the uppermost layer of the DB site (Extended Data Table 1). These ranges of estimates have also been considered in extrapolating to Indonesia's peatlands.

The error ranges of cumulative C stocks have been analysed based on standard deviations of the measured bulk density and C concentration from five replicated samples (Extended Data Table 1). These ranges, now displayed in Fig. 1b, are mostly within the plots and significantly lower than the standard errors evaluated above.

Peat depth lost due to fires was evaluated based on the difference in peat depth (cm) as a function of ^{14}C age (kyr BP) between the DF and DB sites (Extended Data Fig. 5). For the DF site, the ^{14}C age exhibited a significant linear relationship with peat depth (Fig. 1a; $r = 0.98$, $p < 0.001$), and consequently, the peat depth as a function of ^{14}C age was estimated with standard error. The peat depth corresponding to a ^{14}C age of 2.89 kyr, the ^{14}C age of peat in the uppermost layer of the DB site, was then estimated as 42 cm (range: 36–48 cm).

Detailed methods for estimating these values are presented in Methods section, and Supplementary Method 2 and 3 in Supplementary Information. We have also displayed 95% confidence intervals in Figs. 1 and 2.

3. Insufficient site characterization and methodological clarity

Critical details about study sites such as peat thickness, drainage intensity, ditch spacing, vegetation composition are missing or vague. Sampling strategy and collection is unclear (e.g., depth selection, number of samples per site, rationale for layer selection). These gaps make it difficult to assess the validity and comparability of the data.

Response: Detailed descriptions of the study sites have been added as Supplementary Method 1 in Supplementary Information. Key features, including peat thickness, drainage intensity (canal size), vegetation composition, of each site have been summarized and presented in Extended Data Table 2.

In this study, we collected peat samples from the face of a pit excavated using a pit excavation method, rather than from the peat surface using standard core sampling. This approach was specifically chosen to prevent contamination of deeper peat samples (which are largely depleted in ^{14}C due to radioactive decay) by shallower samples (rich in modern C and high in ^{14}C). Contamination during core sampling is a common issue that can significantly affect ^{14}C age determination. However, the pit excavation method we used restricted the depth of peat samples we could collect. We made every effort to conduct peat sampling in September 2014, considering the generally lowest groundwater levels of the year (as described in the Methods section) and the unusually low groundwater levels associated with the 2014 El Niño drought (mean GWL = -0.23 m compared to the normal years' GWL = -0.08 m, as described in the main text) in our long-term observation over 15 years. Deeper peat layers were almost permanently flooded at all sites and were considered well-preserved, with minimal influence from drainage or fires. Radiocarbon analysis was conducted only for selected depth intervals. This decision was based on confirmed linear relationships between cumulative C stock and ^{14}C age (Fig. 1b), as well as between ^{14}C age and depth (Fig. 1a), further supporting the assumption that the peat profiles are well-preserved across all three sites.

A key limitation of this study is the lack of spatial replications within the site, as explicitly acknowledged in the final paragraph of the main text. Expanding our approach to a broader set of peatland ecosystems will allow for a more comprehensive assessment of disturbance impacts, accounting for spatial uncertainties. Currently, to our knowledge, no published data exists on spatial variability in ^{14}C age profiles in tropical peatlands. The following figures present our unpublished data collected at two locations within a peatland (oil palm) in Sarawak, northwest Borneo, Malaysia. These figures show similar ^{14}C age–depth profiles and a consistent relationship between cumulative C stock and ^{14}C age between the two locations, despite a distance of over 0.5 km. This consistency may support the hypothesis of relatively spatially homogeneous peat accumulation at least within such spatial scales in peatland sites.

Fig. ^{14}C age depth profiles of peat (a) and cumulative peat C stocks plotted against ^{14}C age (b) for two peatland sites (shown as red and blue circles) in Sarawak, Malaysia.

4. Literature integration and discussion

The discussion does not adequately engage with existing studies on tropical peatland C dynamics, drainage and fire impacts. Several relevant references are missing, and the manuscript underestimates prior research on DOC and C loss processes.

Response: We appreciate your suggestion to include additional literature in the discussion. As addressed in our responses to your specific comments, we have incorporated several of the references you recommended, particularly those related to tropical peatlands, and additional references. These additions have helped make our discussion more comprehensive.

As you noted, several studies have reported changes in DOC dynamics associated with drainage and fires. We have attempted to incorporate these suggested references into our discussion; however, only a limited number of such studies are available for tropical peatlands.

Our study sites have been extensively investigated over several decades using a wide range of methodologies to examine various ecological processes. This long-term body of research provides a strong foundation for validating the findings derived from our innovative approach presented in this manuscript. We have attempted to interpret and

contextualise our results in light of this substantial prior work. However, ^{14}C age data, particularly regarding DOC and C loss processes, remain largely unavailable.

5. Terminology and consistency issues

Key terms (e.g., 'disturbance gradients,' 'release,' 'immense C reservoir') are vague or ambiguous. Acronyms are inconsistently used, and units are not standardized throughout the text. Expressions like "vast quantities' or "unsustainable' need to be replaced with precise, quantitative language.

Response: Following the advice, we have replaced the terms with precise language. Please also see the responses to the specific comments below.

6. Extrapolation and interpretation

Statements about C release (e.g., 1.38 Gt C) lack clarity on whether they refer to gross emissions, net flux, or cumulative losses. The basis for extrapolation is unclear, and the manuscript does not clearly distinguish between undrained and drained peatlands in its analysis, nor are the expressions sufficiently clear.

Response: We have incorporated detailed descriptions of our estimation methods into the Methods section and Supplementary Method and have recalculated the values with clearer explanations, assumptions, and ranges of estimates. Extended Data Table 5 has been refined for improved clarity and understandability. Consequently, the main text has also been revised to reflect these updates with sufficient detail.

7. Writing quality and clarity

Numerous sentences are unclear, overly complex, or logically inconsistent. The abstract and key paragraphs need restructuring for readability and scientific precision.

Response: Thanks to your valuable and detailed guidance given here, we were able to revise the manuscript, particularly the abstract and key paragraphs, for improved readability and scientific precision.

Specific Comments

- (1) Line 1: Consider a more informative topic, such as 'Progressive release of long-stored carbon due to disturbances in tropical peatlands'**

Response: Following the suggestion, we have revised the title to: "Progressive release of long-stored carbon from tropical peatland disturbances". Thank you.

- (2) Lines 23–27: The sentence is unclear. Consider rephrasing for clarity, for example: 'During 18 years, drainage and subsequent peat fires released approximately 36 kg C m⁻² yr⁻¹, of which drainage accounted for 8 kg C and fire events for approximately 28 kg C.'**

Response: We have revised the text based on your feedback, as follows. However, the suggested revision omitted crucial carbon dating information, a key finding of this research. This has been addressed in the following text for clarification, resulting in a more comprehensive and clearer statement.

(Lines 23–28)

"Our findings reveal that during 18 years (1996–2014), drainage and subsequent peat fires released approximately 30–41 kg C m⁻². Drainage accounted for 5–11 kg C m⁻², primarily from centuries- to millennium-old, previously waterlogged peat. Fire events released approximately 23–32 kg C m⁻² of peat carbon that had accumulated over the past 3,000 years from the upper 0.4–0.5 m of peat, initiating a progressive oxidative decomposition of the older peat."

- (3) Lines 27–29: The phrase 'disturbance gradients' is vague. Specify the types of disturbance (e.g., logging, drainage, fire). The reference to 'from the present to more than 4000 yrs BP' suggests a temporal range, but the sentence implies a spatial gradient linked to disturbance. Clarify whether you mean spatial variation or time progression. Suggest stating that disturbances mobilize older peat C.**

Response: Following the suggestion, we have revised the text to: (Lines 28–30) "The radiocarbon ages of dissolved organic carbon demonstrated that disturbances from drainage and fires mobilised peat carbon preserved up to 4,000 years BP."

- (4) Line 29: Clarify here what is meant by 'Indonesia's peatlands'. Do you refer to drained peatlands, peat swamp forests (PSFs), or all peatlands including swamps? Note that not all swamps are peatlands. This distinction is important for**

interpreting potential C release. I notice that some of the information is provided in 'Extended Data Table 4,' but essential and important details should also be included in the main manuscript so that the context and magnitude are clear to the reader.

Response: We apologise again for the insufficient description. We have revised it for clarity to: (Lines 30–31) “Extrapolation to Indonesia’s drained and fire-damaged peatland areas”.

(5) Lines 29–32: The phrase 'a potential release of 1.38 Gt C between 1996 and 2014' is ambiguous. Does this refer to net emissions (emissions minus sequestration), gross emissions, or cumulative losses? Specify the source of release (peat soils?) and the basis for extrapolation (flux measurements, land-use change data, or model outputs). Include uncertainty estimates—confidence intervals or assumptions are standard for extrapolations.

Response: We have revised the text to explicitly clarify the carbon emission source and the basis for extrapolation, as follows. The methodology underpinning this extrapolation is detailed in the relevant sections of the main text and further elaborated in the Methods section.

(Lines 30–34)

“Extrapolation to Indonesia’s drained and fire-damaged peatland areas suggests a potential release of 0.81–3.70 Gt of peat carbon from disturbances between 1996 and 2014. Ongoing oxidative decomposition of ancient peat in drained peatlands contributes an additional 0.03–0.08 Gt C annually—an accelerating impact on the global terrestrial carbon balance.”

(6) Line 36: The term 'immense C reservoir' is vague. Use quantitative language or comparisons, e.g., 'a reservoir containing X Gt of C.'

Response: Since the quantitative value (105 Pg of C) has already been stated in the previous sentence, we have changed “this immense C reservoir” to “this C reservoir” (Line 38).

(7) Line 40: Rephrase “harbours the largest swath of these ecosystems’ to something clearer, such as ‘contains the largest area of these ecosystems.’

Response: Following the suggestion, we have revised the sentence to: (Lines 42–43) “contains the largest area (25 Mha)⁵ of these ecosystems”, with a quantitative value and its reference.

(8) Line 47: Specify the total area when stating ‘less than 40% of native PSFs remaining undisturbed.’

Response: We have provided the area (Line 49).

(9) Lines 54–55: Clarify ‘28-fold greater than predisturbance rates of loss.’ Greater than what baseline? Provide context.

Response: We are sorry that the sentence was incorrect. Now, we have revised it to: (Lines 56–58) “In western Indonesia, annual C losses due to drainage and fires are estimated to be 28-fold greater than predisturbance rates of carbon uptake²³.”

(10) Lines 55–57: Clarify whether ‘mobilized from deeper peat layers’ refers to C loss from drained and burned areas.

Response: We have revised it to: (Lines 59–60) “lost from deeper peat layers in drained and burnt areas”.

(11) Line 57–59: The phrase ‘previously unexplored C pool’ is unclear. If you mean C loss from deeper peat layers (CO₂, DOC) not fully considered before, state that explicitly.

Response: We have revised it to: (Line 63) “previously preserved peat C pool”.

(12) Lines 60–63: The claim that comprehensive assessments are lacking seems overstated. Several studies have examined drainage and wildfire impacts on tropical peatlands. Revise and cite relevant literature. Some examples are given below (see comments for lines 167–176).

Response: Thank you again for the insightful comment. Although several studies have reported the impacts of drainage and fires on tropical peatlands, we consider that

comprehensive assessments of the “magnitude” and “age” of C loss throughout the “entire sequence of disturbances” from undrained peatlands to drained and subsequent fire impacts are still lacking. In response to the comment, we have revised the sentence as follows.

(Lines 65–68)

“Although there are studies examining drainage and wildfire impacts on tropical peatlands^{1,26-28}, comprehensive assessments of the magnitude and age of C loss throughout the entire sequence of disturbances—from undrained peatlands to drainage and subsequent fire impacts—are still critically lacking.”

(13) Lines 66–67: Avoid nested acronyms. Use simpler acronyms like UPSF and DPSF for undrained and drained PSF consistently.

Response: According to this and other comments, we have revised them for consistency.

(14) Line 67: Clarify “edaphically similar area’ in relation to your research sites.

Response: We have revised the sentence, as follows.

(Lines 71–78)

“We examine three peatland sites that represent a sequence of disturbance within an edaphically similar area: an undrained swamp forest (UF), a drained forest (DF), and a drained, repeatedly burnt ex-forest (DB) (Table 1), and characterize peat C as a cumulative stock function of ¹⁴C age (Fig. 1, Extended Data Table 1) rather than peat depth. These three sites are located within 15 km on flat terrain around the edge of peat domes, originally exhibiting similar vegetation, peat thickness, and depth profiles of dry bulk density, C concentration, and ¹⁴C age of the peat (Table 1, Fig. 1, Supplementary Method 1, Extended Data Tables 1 and 2).”

The detailed descriptions and characteristics of the study sites have been provided as Supplementary Method 1 and Extended Data Table 2.

(15) Line 69: Use ‘peat thickness’ instead of ‘peat depth’ throughout the manuscript.

Response: We use “peat thickness” to indicate the thickness of peat, but “peat depth” is kept to specifically indicate a specific depth or layer, for example to indicate the relationship between ^{14}C age and depth.

(16) Line 74: Avoid vague phrases like ‘this vital C reservoir.’ Use precise language.

Response: We have revised it to: “the peat C reservoir”.

(17) Lines 72–85: The paragraph is unclear and difficult to follow, similar to the abstract. Restructure for clarity, including total C loss and its components. For example: ‘During 18 years, drainage and subsequent peat fires released approximately $36 \text{ kg C m}^{-2} \text{ yr}^{-1}$, of which drainage accounted for 8 kg C and fire events for approximately 28 kg C .’

Response: This paragraph (Lines 80–97), along with the following three paragraphs, focuses on the drainage effects on C dynamics, as introduced by the first sentence: “The drainage of peat swamp forests lowers the groundwater level (GWL), increases exposure of peat to oxygen, and stimulates aerobic microbial activity, thereby accelerating the decomposition of the peat C reservoir^{1,14,15}”. Therefore, the structure of this paragraph is best left as is. However, considering the suggestion, we have revised the sentence for improved readability.

(18) Lines 83–85: The statement on CO_2 emissions aligns with previous GHG balance studies of drained peatlands. Acknowledge prior research and cite references.

Response: Thank you for the suggestion. We have cited the references: Hooijer et al., 2012; Sundari et al., 2012; Itoh et al., 2017; Wijedasa et al., 2018; and Cooper et al., 2020.

(19) Lines 107–113: There is a logical conflict with lines 95–99. The text should clarify that elevated emissions at UF in 2014 (likely due to drought) reduce the observed difference between UF and DF, leading to an underestimation of the drainage effect. The measured difference ($6 \text{ kg C m}^{-2} \text{ yr}^{-1}$) is smaller than it would be under normal conditions; therefore, the true drainage effect (the difference between undrained and drained sites under typical conditions) is larger than $6 \text{ kg C m}^{-2} \text{ yr}^{-1}$ over 18 years.

Response: Considering the comment, we have revised the sentences to: (Lines 121–124) “The elevated emissions at UF in 2014, likely due to drought conditions, reduce the observed difference between UF and DF, resulting in an underestimation of the drainage effect under normal conditions. Therefore, the drainage-induced CO₂ emissions over 18 years would be larger than the estimated value of 1.4 kg C m⁻².”

(20) Line 114: Clarify ‘potentially greater peat-C losses.’ Greater than what?

Response: We have revised it for clarity to: (Lines 128–129) “potentially greater peat-C losses from drained tropical peatlands than our estimate⁴²⁻⁴⁴”.

(21) Line 124–125: Explain what is meant by ‘full extent of their impacts on C dynamics remains largely unknown.’ Beyond CO₂ emissions, what aspects are unknown?

Response: We have revised it for clarity to: (Lines 139–140) “full extent of their impacts on C dynamics in tropical peatlands, including the ages of C released and postfire consequences, remains largely unknown”.

(22) Line 125 and throughout: Ensure consistency in units (e.g., use m⁻² instead of “per square meter”).

Response: We have revised it throughout the text.

(23) Lines 125–126: Clarify. I guess you mean simply that 28 kg C m⁻² was lost from the upper 0.4 m of the DPSF.

Response: We have revised it to: (Lines 141–142) “23–32 kg m⁻² of peat C was lost from the upper 0.4–0.5 m of the peat profile at the DB site”.

(24) Line 135: Rephrase ‘fire events liberated peat C’ to ‘fire events released peat C to the atmosphere.’

Response: We have revised it to: (Lines 150–151) “fire events released peat C accumulated over nearly 3,000 years (Fig. 1) into the atmosphere”.

(25) Lines 143–147: This part is hard to follow and has too many nested ideas.

Thermal analysis → abundance of labile compounds → example (cellulose) → no signal of flame-retardant materials → susceptibility → ancient age. Simplify and clarify. Replace “flame-retardant organic materials’ with ‘highly recalcitrant compounds (e.g., lignin, aromatic structures).’ Explain the link between age and susceptibility more clearly.

Response: We have made the suggested change and revised the text to better explain the link between ^{14}C age and susceptibility, as follows.

(Lines 163–166)

“This finding suggests that the upper peat at the DB site remains highly susceptible to oxidative decomposition, even with its ancient age, although a significant relationship between the relative abundance of labile organic compounds and ^{14}C age of peat was observed across all sites and depths (Fig. 2).”

(26) Lines 151–154: Clarify whether Lupascu et al.’s sites differ from yours. Provide context for comparing surface layers.

Response: At the intact and burnt sites of Lupascu et al. (2020), the top 0–10 cm layers showed ^{14}C ages of modern and approximately 100 years BP, respectively. Because ^{14}C ages of the peat were only reported for the top 0–10 cm and the bottom layer (90–160 cm), the detailed ^{14}C age profiles were unknown. Given this information, it is difficult to provide a detailed comparison with our study, but we may be able to suggest that the peat profiles had a significant thickness of modern peat in the upper layers of pre-fire profiles, or that the surface peat layers lost by fires were shallow despite the burnt site experiencing seven fires during 1998–2016. We have included this discussion, as follows.

(Lines 172–175)

“this discrepancy is likely due to the relatively young (approximately < 100 years BP) upper peat layer at their burnt site. This suggests either a thick modern peat layer in the pre-fire profile, or the loss of only shallow surface peat layers resulting from seven fires that occurred between 1998 and 2016.”

(27) Line 159: State the mean GWL explicitly and reference Table 1.

Response: We have explicitly stated the mean GWL (-0.15 ± 0.13 m), referencing Table 1 (Line 180).

(28) Lines 167–176: Revise this paragraph. Prior studies have examined drainage and fire effects on DOC. Cite relevant references and compare findings in the discussion. Some examples:

Yang, P. et al. (2025). Heating-Induced Redox Property Dynamics of Peat Soil Dissolved Organic Matter in a Simulated Peat Fire: Electron Exchange Capacity and Molecular Characteristics. *Environmental Science & Technology*, 59, 489–500.

Sundari, S. (2020). “Dissolved organic carbon and physicochemical variables of peat water in tropical peat swamp forests.’ *IOP Conference Series: Earth and Environmental Science*, 591, 012045.

Sazawa, K., Wakimoto, T., Fukushima, M., Yustiawati, Y., Syawal, M. S., Hata, N., Taguchi, S., Tanaka, S., Tanaka, D., & Kuramitz, H. (2018). Impact of peat fire on the soil and the export of dissolved organic carbon in tropical peat soil, Central Kalimantan, Indonesia. *ACS Earth and Space Chemistry*, 2(7), 692–701.
<https://doi.org/10.1021/acsearthspacechem.8b00018>

Lupascu, M., Akhtar, H., Smith, T. E. L., Sukri, R. S. (2020). Post-fire carbon dynamics in the tropical peat swamp forests of Brunei reveal long-term elevated CH₄ flux. *Global Change Biology*, 26(9), 5125–5145.

Burd, K., Estop-Aragónés, C., Tank, S. E., Olefeldt, D. (2020). Lability of dissolved organic carbon from boreal peatlands: interactions between permafrost thaw, wildfire, and season. *Canadian Journal of Soil Science*, 100(4), 503–515.

Response: We have revised the first sentence to clarify the main focus of this paragraph, with citation of relevant references, as follows:

(Lines 189–192)

“Crucially, this is the first study to show ¹⁴C ages of DOC across the disturbance sequence, which provides new insights into the impact of peat fires on DOC source and dynamics, extending beyond previous reports that described changes in the concentration and quality of DOC due to drainage and fires^{25,49,51,52}.”

Thank you for your helpful suggestion regarding the literature on DOC dynamics in peatlands. While we consider all of the recommended references valuable, some are not directly relevant to the main focus of our study. Here, we have included a citation to Sazawa et al. (2018), as suggested, which reports laboratory extractions of

water-extractable organic carbon from subsurface peat (30–50 cm depth) at both unburnt and burnt sites. We have also cited Gandois et al. (2013), Evans et al. (2014), and Lupascu et al. (2020), which investigated changes in the concentration and quality of DOC due to drainage and fires. However, we decided not to cite Yang et al. (2025; peat in Florida, USA) and Burd et al. (2020; boreal peat) because the peat at their study sites is likely to have different characteristics from tropical peatlands in Southeast Asia. We also did not include Sundari (2020) because it reports DOC concentrations at our three sites but does not include information on the sampling dates.

Due to the limited availability of published ^{14}C data, we have instead included a sentence to strengthen the C dynamics linking DOC with peat-C within our study, as follows.

(Lines 192–196)

“Radiocarbon data from groundwater at the repeatedly burnt (DB) site reveal that DOC originates from the oxidative decomposition of peat preserved for up to 4,000 years, which is significantly older than the DOC found at the undrained (UF) and drained (DF) sites (Fig. 3) and aligns with the peat-C ages at the DB site (Fig. 1a).”

(29) Line 169: Clarify 'fire-affected (DB) site.' Is this the repeatedly burnt drained site? Be consistent.

Response: We have revised it to: (Line 193) “repeatedly burnt (DB) site” for consistency.

(30) Line 171: Clarify 'severely drained (DF) sites.' Maintain consistent terminology.

Response: We have revised it to: (Line 195) “undrained (UF) and drained (DF) sites” for consistency.

(31) Line 178: Avoid vague phrases like 'sequestered vast quantities of C.' Provide numbers.

Response: We have revised it as suggested (Line 202).

(32) Lines 178–183: 'readily reversed'. Does this mean rapid C loss or degradation, or both or something else?). Replace vague language with precise statements. You

could say for example: 'Our results show that peat C is highly vulnerable to loss under human-induced disturbances such as drainage and fire, indicating that its long-term stability cannot be guaranteed under current land-use practices.'

Response: Thank you for your guidance. We have revised the sentences accordingly, as follows.

(Lines 203–205)

“our results show that peat C is highly vulnerable to loss under human-induced disturbances such as drainage and fire, indicating that its long-term stability cannot be guaranteed under current land-use practices.”

(33) Line 180: Throughout the ms, ensure consistency in terms like 'C storage,' 'stocks,' and 'store.'

Response: We now consistently use “stock” throughout the manuscript.

(34) Lines 182–183: Clarify whether recovery is likely/unlikely under changing climate conditions.

Response: Thank you for your insightful comment. We have included the following discussion.

(Lines 207–212)

“Reclaiming drainage channels is considered a strategy to reduce C emissions and recover peat C stock⁵⁴. However, fully restoring the original forest cover, particularly in burnt areas, is challenging due to a lack of seeds and nutrients⁵⁴, and postfire flooding caused by land subsidence from peat loss¹ (Table 1). Projected increases in temperature and GWL in this region⁵⁵ further complicate the outcome of peatland C recovery under a changing climate.

(35) Line 184: Specify whether 'from our PSF' refers to drained sites, repeatedly burnt sites or both.

Response: We have revised it to: (Lines 213–214) “from both our drained and repeatedly burnt sites”.

(36) Lines 187–189: Clarify whether extrapolation applies to all peatlands. Separate undrained and drained peatlands in analysis. See my earlier comment about Indonesia’s peatlands.

Response: We apologise again for the insufficient descriptions regarding the extrapolation. We have already applied our results of drainage- and fire-induced C losses separately to drained and fire-damaged peatland areas of Indonesia, respectively (Extended Data 5). Considering this and earlier comments, we have revised the text to: (Lines 216–217) “By extrapolating our results to drained and fire-damaged peatland areas of Indonesia (Extended Data Table 5), we estimate...”. We have also incorporated detailed descriptions of the estimation methods into the Methods section and Supplementary Method, and recalculated the values with ranges of estimates (Extended Data Table 5) to improved clarity.

(37) Line 192: Provide the area of drained peatlands.

Response: We have provided the area of drained peatlands (Line 224).

(38) Lines 195–199: A sensitivity analysis is essential due to high uncertainty and small sample size. Use error ranges (e.g. NEE-values of Table 1) and compare bulk density values with other studies. Include references for dry bulk density and C concentrations.

Response: Thank you for the guidance. We have analysed the sensitivity of estimates in both our experimental results and extrapolation. All estimates are now provided with ranges. Please also see the response to major comment 2. Data on dry bulk density and C concentrations for our peat samples are available and have already been presented in Extended Data Table 1.

(39) References: Ensure uniform reference style.

Response: We checked again the manuscript’s reference style. It has also been checked by a professional editing and formatting service by Springer Nature Author Services.

(40) Figure 1: Add GWL range for clarity.

Response: We have indicated the GWL range in Fig. 1a, referencing Table 1.

(41) Lines 356–362: Provide details on ditching and spacing at study sites. How are the sites comparable to drained sites of Indonesia? Define 'severely drained forest (DF).'

Response: We have provided information about ditching at the study sites in the Methods section, Supplementary Method 1, and Extended Data Table 2, and have revised “a severely drained forest (DF)” to “a drained forest (DF)” for consistency (Line 423).

It is difficult to say whether the drainage conditions are similar to those in other regions. However, the average groundwater level difference between UF and DF is 0.32 m (Hirano et al., 2024), which is nearly the same as the 0.33 m average groundwater level difference between undrained and drained forests in the study area (Sumatra, Borneo, and the peninsula Malaysia) over a 10-year period (Hirano et al. 2025). Therefore, the drainage intensity of DF can be considered approximately equivalent to that of drained forests in the tropical peatlands of Southeast Asia. This information has been included in the Methods section.

DB is somewhat more complicated. While it is drained similar to DF by canal excavation, sparse vegetation caused by surface peat fires results in lower water loss through transpiration, leading to groundwater levels that are approximately the same as those of the undrained UF (Table 1, Hirano et al., 2012). Subsidence caused by surface peat fires also likely contributes to these relatively high groundwater levels at DB.

(42) Line 362: Include key site characteristics (e.g. size, peat thickness) in Table 1 for context. At the moment, there is very little info of the sites, only a single reference to Hirano et. al.

Response: We have included peat thickness in Table 1. In addition, detailed descriptions of the study sites are provided in Supplementary Method 1, and key site characteristics including vegetation, drainage intensity, and microtopography, are summarized in Extended Data Table 2.

(43) Line 366: Replace vague 'rich shrubbery understory' with species names.

Lines 365–367: Please, be more specific. Specify dominant native tree species’

Response: We have revised the sentence for clarification to: (Lines 432–434) “The UF site is a swamp forest dominated by *Combretocarpus rotundatus* and *Cratoxylum arborescens*, with a rich understory of shrubs largely composed of their sapling⁵⁸⁻⁶⁰.”

(44) Lines 367–368: Clarify ‘a national park since at least 2006.’

Response: We have revised the sentence for clarification to: (Lines 434–435) “The site was in a National Park designated in 2006”.

(45) Lines 367–368: ‘formerly a PSF’? Aren’t all sites PSFs with different disturbance histories?

Response: We have revised the sentences for clarification as follows.
(Lines 436–439)

“The DF and DB sites, formerly undrained swamp forest, have been drained by canal excavation since 1996 (ref.⁵⁴). The DB site experienced repeated fires in 1997, 2002, 2009, and 2014, i.e., El Niño years, resulting in the repeated loss of surface peat and vegetation (shrubs, ferns, and grasses).”

(46) Lines 378–381: ‘Selected layers’. Explain how layers were selected (visual characteristics, stratigraphy, C content, something else). State sampling depth range and number of samples per site. Site-specific variation: Were the same depths sampled at all sites? Does the sampling begin at the surface and go down to a specific depth or only certain horizons were targeted? Note that the sample size (three profiles, 15 samples) is very small and lacks spatial coverage, limiting generalization.

Response: This study focuses on assessing drainage and fire impacts on C dynamics. To achieve this, we primarily targeted peat layers located above the groundwater level. As mentioned in the response to major comment 3, we collected peat samples using a pit excavation method rather than core sampling method to prevent contamination of peat samples in the depth direction. Consequently, the maximum sampling depth varied depending on the site (i.e., groundwater levels). Although selected sampling layers differed between sites, the depth profiles of dry bulk density and C concentration were

nearly constant from the surface to the bottom of the sampling layer and among the five replicated samples within each layer, at each site (as presented in Extended Data Table 1). In addition, at all three sites, ^{14}C age increased almost linearly with depth (Fig. 1a), indicating well-preserved, stratified profiles from the surface to the bottom. This allows for an estimation of C stocks in the profile based on the results from the selected sampling layers. Considering these points, we have revised the sentences as follows.

(Lines 446–455)

“Peat sampling was conducted via the pit excavation method at all three sites to prevent contamination of samples in the depth direction. Sampling was performed in September 2014, considering the generally lowest GWLs of the year and the unusually low GWLs associated with the 2014 El Niño drought²¹. Briefly, a soil pit was dug at least 1 m apart from any trees at each site, and peat samples were collected from selected layers at 5-cm depth intervals using a 100-cm³ stainless steel cylinder. The maximum sampling depth varied among the sites (45 cm, 75 cm, and 45 cm for the UF, DF, and DB sites, respectively), depending on the GWL. Although the sample size was small (17 for three profiles), the results indicate well-preserved, stratified profiles from the surface to the bottom for each site. The sampling did not provide spatial coverage for each site.”

(47) Line 383: 'dry bulk density.

Response: Revised.

(48) Line 389: Specify whether subsamples or whole samples were used. Sample size?

Response: We have included the information as: (Lines 461–462) “Radiocarbon analysis was conducted on subsamples (equivalent to approximately 4 mg of C)”.

(49) Lines 401–402: Clarify whether TG-DTA was used to identify labile and recalcitrant compounds as two groups or within each category.

Response: The revised categorisation is now clearly stated as: (Line 478) “To differentiate labile and recalcitrant organic compound groups”.

(50) Line 417: State how long samples were stored at 4°C before analysis.

Response: We have revised the sentences, including information about sample storage, as follows.

(Lines 493–496)

“The groundwater samples were subsequently stored at 4°C under low pH conditions until freeze-drying. In October 2020, DOC samples were obtained as powdered samples by freeze-drying approximately 60–250 ml of the stored samples after confirming no changes in DOC concentration during storage.”

(51) Lines 418–419: Clarify what is meant by ‘some samples.’ How many? Were all samples analyzed?

Response: We analysed two additional samples that were immediately freeze-dried without storage in liquid state. We have revised the sentences for clarity as follows.

(Lines 497–501)

“For two samples, an aliquot was taken immediately after the DOC concentration was measured, and DOC samples were obtained by freeze-drying and stored as powdered samples in sealed containers until 2020. These two samples were also used for confirmation of storage effect. AMS-¹⁴C analysis was performed on subsamples (equivalent to approximately 1 mg of C) from both sets of DOC samples.”

(52) Lines 430–431: Yes, incomplete sampling and small sample size are major limitations. This issue underlines the need for sensitivity analysis.

Response: We have included the sensitivity analysis and now provided the ranges of estimates in the revised manuscript. Please also see the response to the major comment 2.

Reviewer #2 (Remarks to the Author):

(1) Line 44; Any justification for using the term PSF vs. peatland? The paper starts by using peatland, then switches to PSF, and later returns to peatland again. Preferably, just use the term peatland consistently.

Response: Thank you for the valuable comment. Now, we use the term “peatland” consistently throughout the manuscript.

(2) Line 66; Please revise the sentence to specify that UF, DF, and DB are peatland sites where the samples were collected. Currently, it sounds like they are three types of PSFs.

Response: Agreed. The revised specifications are now clearly stated as: (Lines 72–73) “an undrained swamp forest (UF), a drained forest (DF), and a drained, repeatedly burnt ex-forest (DB)”.

(3) Line 385; Is there any radiocarbon date calibration? It needs to be specified in the methods section: what software was used to calibrate the dates, which calibration curve was used, etc. It is also preferable to calibrate the radiocarbon data and present the ages in cal BP instead of kyr BP (uncalibrated).

Response: Since precise dating is not required for the purpose of this study, and to facilitate comparison with uncalibrated ^{14}C ages in previous literature, we report conventional ^{14}C ages. However, we also present calibrated ages in Extended Data Tables 6 and 7. Calibration methods have been included in the Methods section (Lines 470–472).

(4) Fig. 1a; Instead of using linear age, can you conduct an age–depth analysis (e.g., Bacon) and integrate the results into this graph?

Response: Peatlands are well-preserved systems, allowing for a very good fit with linear regression in age–depth analysis (Fig. 1a). However, when applying the proposed method to typical soils where organic matter is rapidly decomposed and less accumulated in deeper layers, it will likely be necessary to consider alternative approaches for characterizing age–depth relationships, as suggested.

(5) Figs. 1 and 2; Can you please specify on the x-axis that the radiocarbon dates are from peat soil samples (maybe “ ^{14}C age of peat soil”)? You have two sets of dates — one for peat soil and one for DOC. In Fig. 3 it is clearly labeled as DOC dates (“ ^{14}C age of DOC” on the x-axis), but in Figs. 1 and 2 it needs to be specified more clearly.

Response: We have specified the notation on the X-axis in Figs. 1 and 2 to make it clear that it represents the ^{14}C age of the peat, as suggested.

(6) Fig. 4; Great figure, very informative. Can you also add the amount of subsidence from UF to DF? Since you mention that subsidence from DF to DB is 40 cm, it would be better if you also mention the value for UF to DF.

Response: Thank you for your insightful comment. Building on this suggestion, we estimated subsidence from UF to DF based on the estimated loss of peat C from the upper 0–45 cm of the peat profile at UF. Considering that the dry bulk density and C concentration are nearly constant throughout the peat profile at UF (Extended Data Table 1), we could assume that the percentages of peat C loss correspond directly to the percentages of peat volume loss within the peat matrix of the upper 0–45 cm. The estimated subsidence was 9–21 cm. This estimation is discussed briefly in the revised text as follows and is also illustrated in Fig. 4.

(Lines 89–92)

“Given the nearly constant dry bulk densities and C concentrations in the peat profile at UF (Extended Data Table 1), the subsidence caused by this peat loss at DF is estimated at approximately 9–21 cm, with an annual rate of $0.5\text{--}1.2\text{ cm y}^{-1}$. This range aligns with recent estimates for Southeast Asian drained peatlands³¹.”

Reviewer #3 (Remarks to the Author):

General comments

The proposed study investigates the impact of drainage induced aerobic decomposition and fires on carbon gas emissions from tropical peatlands in Indonesia, using the radiocarbon (^{14}C) method. Southeast Asian tropical peatlands represent an exceptionally large but highly threatened carbon storage, which is rapidly losing due to drainage and fires. This research aims to assess changes in peat quantity through an approach that is less dependent on measurement and prediction errors, as well as sample representativeness, issues that often affect conventional soil carbon flux measurement techniques such as so called chamber methods.

Response: Thank you for the thorough review and evaluation of our manuscript.

Radiocarbon analysis is a relevant tool for determining peat age, particularly given that the age range of peat layers falls within the time range where the method is most reliable. The applied approach, comparing age estimates across sites subjected to different treatments (undrained control, drained, and burned), is valid in that it avoids biases in the estimates of mass loss caused by e.g. peat subsidence. However, age observations from the undrained site do not extend very deep in the peat layer, and it would have been interesting to see how the ages of the deeper peat layers are. There is strong reason to assume a linear correlation between peat age and depth (Fig. 1), but this relationship varies among sites; it is clearly steepest at the undrained site. Further clarification of these differences would be valuable; do they result from insufficient observations or from “genuine” site-specific variations? This question is important, as it influences quantitative estimates of carbon losses.

Response: Thank you for the insightful comment, particularly highlighting the innovative aspects of this study.

The limited extent of the ^{14}C age observations from the undrained (UF) site is attributed to our sampling method. We collected peat samples from the face of a pit excavated using a pit excavation method, rather than from the peat surface using standard core sampling. This approach was specifically chosen to prevent contamination of deeper peat samples (which are largely depleted in ^{14}C due to radioactive decay) by shallower samples (rich in modern C and high in ^{14}C). Contamination during core sampling is a common issue that can significantly affect ^{14}C age determination. However, the pit excavation method we used restricted the depth of peat samples we could collect. We made every effort to conduct peat sampling in September 2014, considering the generally lowest groundwater levels of the year (as described in the Methods section) and the unusually low groundwater levels associated with the 2014 El Niño drought (mean GWL = -0.23 m compared to the normal years' GWL = -0.08 m, as described in the main text). Deeper peat layers were almost permanently flooded at all sites and were considered well-preserved, with minimal influence from drainage or fires.

The difference in the slope of the relationship between peat age and depth is currently difficult to assess due to a lack of spatial replication at our sites and no published data existing on spatial variability in ^{14}C age profiles in tropical peatlands. The following figures present our unpublished data collected at two locations within a peatland (oil palm) in Sarawak, northwest Borneo, Malaysia. These figures show similar ^{14}C age–

depth profiles and a consistent relationship between cumulative C stock and ^{14}C age between the two locations, despite a distance of over 0.5 km. This consistency may support the hypothesis of relatively spatially homogeneous peat accumulation at least within such spatial scales in peatland sites.

Fig. ^{14}C age depth profiles of peat (a) and cumulative peat C stocks plotted against ^{14}C age (b) for two peatland sites (shown as red and blue circles) in Sarawak, Malaysia.

To evaluate the ranges of our estimates for drainage- and fire-induced C losses, we have performed sensitivity analysis based on our dataset and reflected the results in the revised manuscript (value notations in the text, Fig. 1, Supplementary Method 2, and Extended Data Table 5). Carbon losses due to disturbances have been evaluated based on differences in cumulative C stocks (kg C m⁻²) as a function of ^{14}C age (kyr BP) between the two sites of interest (UF and DF for drainage-induced C loss and DF and DB for fire-induced C loss). For all sites, the cumulative C stocks exhibited significant linear relationships with ^{14}C age (Fig. 1b; UF: $r = 0.99$, $p < 0.05$, DF: $r = 0.99$, $p < 0.0005$, DB: $r = 0.97$; $p < 0.01$) and were estimated as a function of ^{14}C age, with standard errors. Using these relationships and standard errors, the drainage-induced loss of C accumulated over the past 1,000 years was estimated as 8.0 kg C m⁻² (range: 4.9–11.1 kg C m⁻²) as the difference in cumulative C stocks between DF and UF (UF minus DF) at a ^{14}C age of 1 kyr BP. Similarly, the fire-induced C loss was estimated as 27.6 kg C m⁻² (range: 22.8–32.5 kg C m⁻²) as the difference in cumulative C stocks between DF and DB (DF minus DB) at a ^{14}C age of 2.89 kyr BP, the ^{14}C age of peat in the

uppermost layer of the DB site (Extended Data Table 1). These ranges of estimates have also been considered in extrapolating to Indonesia's peatlands.

Our research emphasizes the need for future research with larger and more spatially diverse datasets, including within-site and between-site variability for a sequence of disturbances. This point has been strengthened in the last sentences of the main text.

An important question is whether the studied sites can truly be considered representative of one another in terms of peat layer characteristics. If this assumption can be credibly justified, the knowledge produced by this study would have significant global relevance and could fundamentally reshape estimates of carbon loss from tropical peatlands, associated risk assessments, and projections of future carbon emissions. At least, the study is a promising step forward in the development of GHG inventory methodologies and introduces a novel assessment approach that could be applied to peatlands worldwide. I recommend acceptance of the manuscript following the above-mentioned revisions.

Response: Thank you for your insightful comment and high evaluation, highlighting the potential significance of this work, with a recommendation for acceptance after certain revisions.

We have included, to the best of our ability, detailed descriptions and characteristics of the study sites as Supplementary Method 1 and Extended Data Table 2. We have also incorporated improvements to the site descriptions in the main text and the Methods section.

These three sites are geographically proximate (Moore et al., 2013), situated within 15 km on relatively flat terrain around the edge of peat domes. They exhibit similar vegetation, peat thickness, and depth profiles of dry bulk density, C concentration, and ¹⁴C age of peat (Table 1, Fig. 1, Supplementary Method 1, Extended Data Tables 1 and 2). This close proximity facilitates a comparative assessment of disturbance impacts on C dynamics within peatlands over several decades (e.g., Page et al., 2002; Hirano et al., 2012, 2024, 2025; Moore et al., 2013; Itoh et al., 2017; Könönen et al., 2018; Ohkubo et al., 2021), including CO₂ release through oxidative peat decomposition, DOC export via channels, drainage impact estimates based on groundwater levels, and fire impact estimates based on ecosystem C exchange. Therefore, a strong point of our study is that

this extensive prior research provides reliable validation for the findings from our innovative approach, as we present in this manuscript. Although we acknowledge the limited sample size and lack of spatial replication, comprehensively assessing this existing knowledge allows for a thorough discussion and assessment of our results. Again, however, further research is needed to provide a robust assessment of the regional impacts of peatland disturbances on the global C cycle.

Detailed comments

r.133: This is quite a drastic result, but the order of magnitude seems to be correct.

Response: This conclusion, led by our results, is also supported by accumulating observations at the study sites.

r.146: It is noteworthy that the exposure of deeper layers to aerobic decomposition did not differ from the decomposition potential of surface peat layers based on chemical characteristics.

Response: This finding aligns with observations at our burnt (DB) site, which show ongoing oxidative peat decomposition (Itoh et al., 2017), as discussed in the following sentence (Lines 166–168).

r.187–189: Here, the emissions from drainage and fires appear to have been extrapolated to the entire peatland area of Indonesia (20.7 M ha)? However, it should be noted that this area includes all peatlands, including those in natural undrained state. The calculation therefore seems overestimated if the assumption is that this entire area has burned and been drained?

Response: We apologise for the insufficient descriptions regarding the extrapolation. We have already applied our results of drainage- and fire-induced C losses separately to drained and fire-damaged peatland areas of Indonesia, respectively (Extended Data Table 5). We have revised the text to: (Lines 216–217) “By extrapolating our results to drained and fire-damaged peatland areas of Indonesia (Extended Data Table 5), we estimate...”. We have also incorporated detailed descriptions of the estimation methods into the Methods section and Supplementary Method, and recalculated the values with ranges of estimates (Extended Data Table 5) to improved clarity. In addition, we have

revised the text in the Abstract to: (Lines 30–31) “Extrapolation to Indonesia’s drained and fire-damaged peatland areas suggests...”.

Table 1: It is remarkable that respiration is not highest at burned and drained sites but at the undrained site, where high GPP compensates for losses. Could a note be added here on the method used to measure respiration?

Response: CO₂ emissions through peat decomposition were measured in 2014. This information has been included in the footnote of Table 1, with an appropriate reference. Measurements were conducted approximately monthly using an infrared gas analyser with a closed chamber method incorporating root-trenching (as briefly described in the main text: Lines 100–103).

r.378: How was the starting level (0-level) for peat sampling defined at the different sites?

Response: Pit locations were selected to be at least 1 m apart from any trees and with minimal coarse woody debris on the surface. If woody debris was present, it was removed before excavating the pit. Peat samples were then collected from the pit face, a method that allowed for easy identification of the peat surface. The maximum depth of the collected samples varied depending on groundwater levels at each site.

r.409–411: The number of water samples taken is very low. It is difficult to assess intra-annual variation in concentrations based on these sampling frequency, especially after fire. Have the DOC concentrations been measured in previous studies with higher resolution that could be cited?

Response: Agreed. Our colleague monitored DOC concentrations at our UF, DF, and DB sites bi-weekly during non-fire year (2011), revealing mean (\pm SE) DOC concentrations of 27.9 ± 4.0 (UF), 71.0 ± 4.7 (DF), 46.3 ± 7.0 mg C L⁻¹ (DB) (Sundari, unpublished data). While these data have not yet been published and cannot be cited in the text, they indicate relatively stable intra-annual DOC concentrations at each site (unless fire occurs).

The primary argument here is that the differences in DOC-¹⁴C age across the three sites, and their link to the ¹⁴C depth profile of the peat, confirm the conclusion that

disturbances contribute to the release of ancient C through both CO₂ emissions and fluvial pathways. Moore et al.'s work (Moore et al., 2013) clarified the impact of logging and drainage on the age of DOC transported through channels. Future research will focus on detailed observations of seasonal and fire-event related fluctuations in DOC concentration within the peat profile and the impact of fires.